# Wnt target gene activation requires β-catenin separation into biomolecular condensates

**Richard A. Stewart, Zhihao Ding, Ung Seop Jeon, Lauren B. Goodman, Jeannine J. Tran, John P. Zientko, Malavika Sabu, Ken M. Cadigan**◯ *

Department of Molecular, Cellular, and Developmental Biology, University of Michigan, Ann Arbor, Michigan, United States of America

* cadigan@umich.edu

**Data Availability Statement:** All relevant data are within the paper and its Supporting Information files.

## Abstract

The Wnt/β-catenin signaling pathway plays numerous essential roles in animal development and tissue/stem cell maintenance. The activation of genes regulated by Wnt/β-catenin signaling requires the nuclear accumulation of β-catenin, a transcriptional co-activator. β-catenin is recruited to many Wnt-regulated enhancers through direct binding to T-cell factor/lymphoid enhancer factor (TCF/LEF) family transcription factors. β-catenin has previously been reported to form phase-separated biomolecular condensates (BMCs), which was implicated as a component of β-catenin's mechanism of action. This function required aromatic amino acid residues in the intrinsically disordered regions (IDRs) at the N- and C-termini of the protein. In this report, we further explore a role for β-catenin BMCs in Wnt target gene regulation. We find that β-catenin BMCs are miscible with LEF1 BMCs in vitro and in cultured cells. We characterized a panel of β-catenin mutants with different combinations of aromatic residue mutations in human cell culture and *Drosophila melanogaster*. Our data support a model in which aromatic residues across both IDRs contribute to BMC formation and signaling activity. Although different Wnt targets have different sensitivities to loss of β-catenin's aromatic residues, the activation of every target examined was compromised by aromatic substitution. These mutants are not defective in nuclear import or co-immunoprecipitation with several β-catenin binding partners. In addition, residues in the N-terminal IDR with no previously known role in signaling are clearly required for the activation of various Wnt readouts. Consistent with this, deletion of the N-terminal IDR results in a loss of signaling activity, which can be rescued by the addition of heterologous IDRs enriched in aromatic residues. Overall, our work supports a model in which the ability of β-catenin to form biomolecular condensates in the nucleus is tightly linked to its function as a transcriptional co-regulator.

## Introduction

The Wnt/β-catenin signaling pathway is evolutionarily conserved across metazoans and is indispensable for organismal development and a variety of adult tissue functions [1,2]. The primary output of this pathway is the differential regulation of gene expression programs,

**Funding:** This work was supported by a grant from the US National Institute of General Medical Sciences grant NIH R01GM108468) to KMC as well as Chair's Research Funds from the College of Literature, Arts and Sciences (LSA) at the University of Michigan (U-M) to KMC. RAS was partially supported by the University of Michigan Predoctoral Training in Genetics (T32 GM007544). JPZ was supported by the U-M Rackham Graduate School Pathway Master's Program. The funders had no role in study design, data collection and analysis, decision to publish, or preparation of the manuscript.

**Competing interests:** The authors have declared that no competing interests exist.

**Abbreviations:** APC, adenomatosis polyposis coli; BMC, biomolecular condensate; BSA, bovine serum albumin; CKI, Casein Kinase I; GSK3, Glucagon Synthase Kinase-3; HRP, horseradish peroxidase; IDR, intrinsically disordered region; IF, immunofluorescence; PFA, paraformaldehyde; PVDF, polyvinylidene fluoride membrane; ROI, region of interest; TCF/LEF, T-cell factor/lymphoid enhancer factor; WRE, Wnt-response element.

which is accomplished through the nuclear accumulation of β-catenin, a transcriptional co-regulator. β-catenin regulates Wnt targets in conjunction with transcription factors, the most prominent of which are members of the T-cell factor/lymphoid enhancer binding factor (TCF/LEF) family. Nuclear β-catenin binds with TCF/LEFs on the chromatin at *cis*-regulatory Wnt-response elements (WREs) [3,4]. Many cancers are causally linked to the inappropriate elevation of nuclear β-catenin, which can occur through loss of function mutations in negative regulators of the pathway, such as adenomatosis polyposis coli (APC), Axin, and Ring finger protein 43 (RNF43), or through oncogenic mutations in β-catenin that prevent its turnover [5].

β-catenin is conventionally understood to be comprised of 3 distinct domains: an intrinsically disordered N-terminal domain (N-IDR), a highly structured internal domain consisting of 12 Armadillo (Arm) repeats, followed by an intrinsically disordered C-terminal domain (C-IDR) [3,6]. β-catenin's N-IDR is necessary for regulating the stability of the protein and contains a region bound by α-catenin [6]. Cytosolic β-catenin is bound by a "destruction complex," which contains APC, Axin, as well as 2 kinases, Casein Kinase I (CKI) and Glucagon Synthase Kinase-3 (GSK3), which serially phosphorylate 4 serine/threonine residues, priming the protein for proteasomal degradation [7]. The Arm repeat region contains binding sites for TCF/LEF transcription factors and E-cadherin (repeats 3–8), as well as co-activators BCL9 and BCL9L (repeat 1) [3,8–10]. The last 4 Arm repeats and the C-IDR are bound by a variety of transcriptional regulators, including chromatin remodelers such as Brg-1 and the histone acetyltransferases CBP/p300 [3,11–13]. The current model suggests that factors are sequentially recruited to WRE chromatin (i.e., TCF/LEFs recruit β-catenin, β-catenin recruits additional co-regulators) and this is sometimes referred to as the "chain of adaptors" model [14]. While there is significant support for TCF/LEFs, β-catenin, and other co-regulators physically interacting with each other to promote transcription, the exact nature of these interactions remains to be determined.

Recent studies indicate a role for biomolecular condensates (BMCs) in transcriptional activation [15–18]. BMCs are dynamic, membraneless assemblies comprised of proteins and, frequently, nucleic acids. Weak, multivalent interactions between the IDRs of constituent proteins drive the formation of BMCs, which is usually thought to occur through a liquid–liquid phase separation mechanism. Functionally, BMCs are thought to affect biochemical reactions by concentrating molecules, which can have potentiating or inhibitory effects [15]. The evidence for BMCs having a role in transcriptional regulation is derived from live imaging studies demonstrating the existence of dynamic puncta at enhancer chromatin and the propensity for many transcriptional regulators to form BMCs in vitro [15,17,19]. However, rigorous evidence for a physiological role for BMCs, e.g., provided by specific mutations in transcriptional regulators is still lacking.

Previous reports have demonstrated that β-catenin protein can form homotypic and heterotypic BMCs in vitro [20–22]. β-catenin's terminal IDRs were necessary and sufficient for BMC formation, and this behavior was dependent on aromatic amino acid residues within the IDRs. Endogenously expressed and transfected GFP-tagged β-catenins were shown to form dynamic puncta in response to Wnt signaling in cultured cells [20–22], and these puncta are associated with active sites of transcription [21]. Consistent with this, a mutant of β-catenin with IDRs lacking aromatic residues (19 total aromatic amino acid substitutions in the terminal IDRs) was defective in regulating Wnt target genes, recruitment to WRE chromatin, and puncta formation [20,21]. In addition, several peptides that bind to β-catenin inhibit β-catenin condensation and transcriptional activity [21].

While the previously published work is consistent with a physiological role for β-catenin condensates, several key issues remain uncertain. For example, it was not clear that condensate-deficient β-catenin accumulates in the nucleus at levels comparable to wild-type β-

catenin. Additionally, addition of β-catenin binding peptides or the alteration of 19 aromatic residues in β-catenin's terminal IDRs may disrupt key protein–protein interactions that are essential for β-catenin function, regardless of a condensate mechanism. These factors are potential explanations for the defect in recruitment to chromatin and transcriptional activation of the β-catenin aromatic mutant or peptide treated cells in those reports [20,21]. Whether the ability of β-catenin to form BMCs is linked to its activity as a transcriptional co-activator requires further investigation.

In this report, we address the hypothesis that BMC formation is important for β-catenin–mediated regulation of Wnt target genes by generating and characterizing a panel of β-catenin mutants utilizing in vitro and in vivo experimental systems. The results support a model in which the aromatic amino acid residues in both the N- and C-IDRs contribute to BMC formation and transcriptional activity. Importantly, we found that the N-IDR of β-catenin has a previously underappreciated role in transcriptional regulation [23]. Supporting these findings, β-catenin was found to efficiently form heterotypic condensates with LEF1 in vitro and in cultured cells, which also depend on IDR aromaticity of both proteins. Transgenic *Drosophila* lines expressing analogous Armadillo (Arm, the fly ortholog of β-catenin) mutants demonstrated the importance of aromatic residues for Arm activity in the context of fly development. Interestingly, while the mutants displayed lower signaling activity, different Wnt targets (in both human cells and *Drosophila*) had different sensitivities to the loss aromatic residues. Finally, heterologous IDRs from proteins with no known role in transcriptional regulation were able to functionally replace the N-IDR of β-catenin, providing compelling evidence that the ability of β-catenin to form biomolecular condensates is inextricably linked to its role as a transcriptional activator of Wnt target genes.

## Results

### Aromatic amino acid residues within β-catenin's terminal IDRs contribute to homotypic and heterotypic condensate formation in vitro

For many proteins, BMC formation is thought to arise from weak, multivalent interactions between protein IDRs [15]. The N- and C-terminal regions of β-catenin are largely predicted to be disordered, illustrated by 2 independent methods of analysis, IUPred2A [24,25] and AlphaFold [26,27] (S1A and S1B Fig). AlphaFold predicts an α-helix in the N-terminal region from amino acids 121–146, which overlaps with the α-catenin binding domain [28]. The rest of the N-terminal domain and the C-terminal domain have no confidently predicted structure. Previous work from Zamudio and colleagues demonstrated that β-catenin can form homotypic BMCs in vitro, and that the terminal IDRs are necessary and sufficient for condensate formation. Additionally, they showed that the aromatic amino acid residues within both IDRs are required for this behavior [20]. However, the contributions of aromatic residues from the individual IDRs were not examined. Given that these regions have distinct roles in β-catenin function, i.e., N-IDR contains phosphorylation sites controlling β-catenin stability and C-IDR is required for co-regulator activity, we were motivated to examine the requirements for the aromatic residues in more detail.

To address the role of aromatics in each IDR of β-catenin, we recombinantly expressed four eGFP-β-catenin fusion proteins: one containing the full set of aromatic residues plus a S33Y point mutation (β-catenin*). This mutation was incorporated for direct comparison with the β-catenin mutants used for subsequent functional studies. Additional proteins have the N-IDR aromatic residues mutated to alanine (aroN; 9 substitutions), the C-IDR aromatics mutated to alanine (aroC; 10 substitutions), and aromatics in both IDRs mutated to alanine (aroNC; 19 substitutions) (Figs 1A and S2 for sequence information). These proteins were tested for

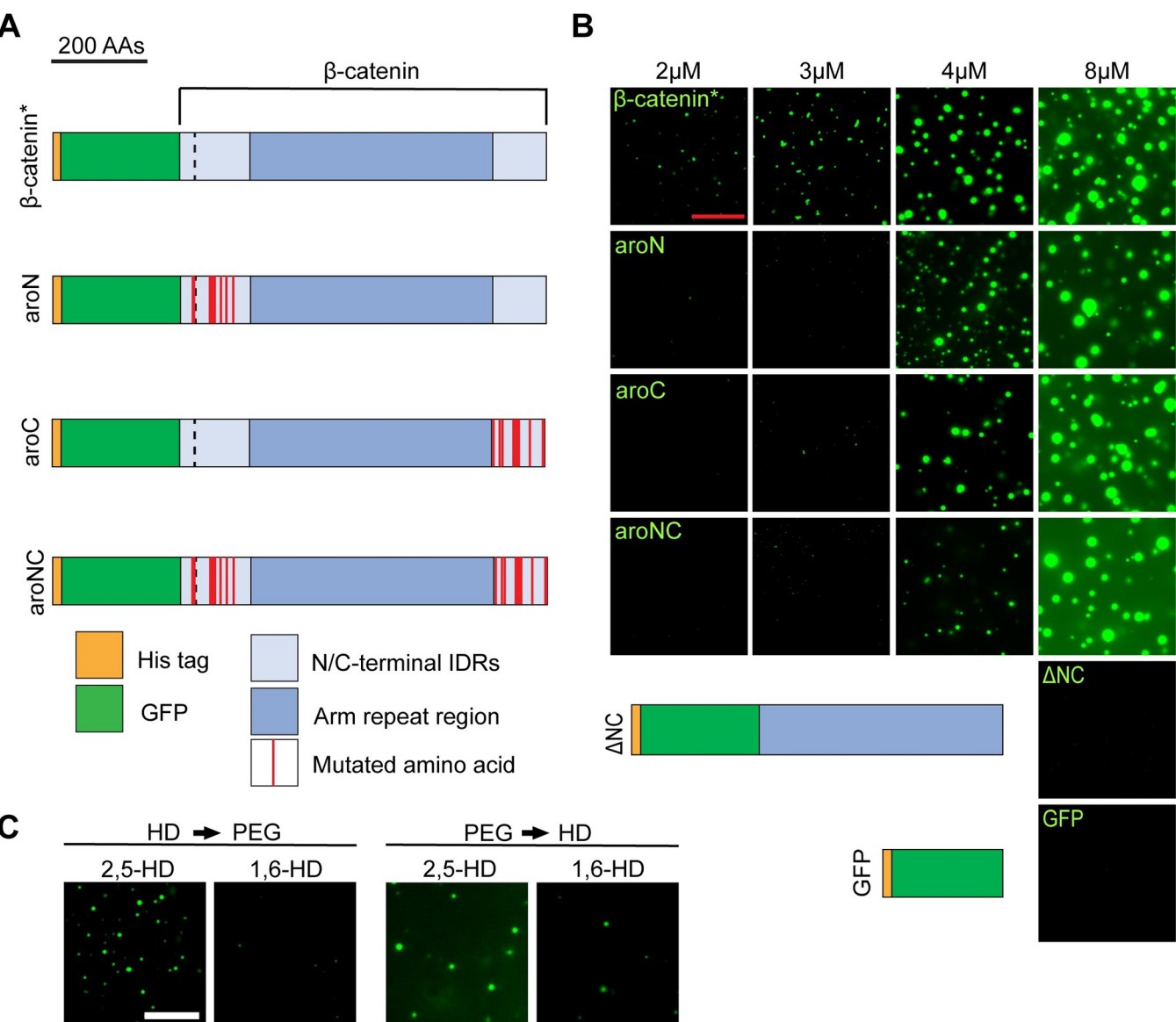

**Fig 1. Aromatic amino acid residues in the terminal IDRs of β-catenin promote biomolecular condensate formation in vitro.** (A) Cartoon representation of the eGFP-β-catenin* protein, the aromatic mutant derivatives, and control constructs. β-catenin* contains one S33Y mutation (dashed black line) in the N-IDR. aroN has all 9 endogenous aromatic amino acids within the N-IDR mutated to alanines. aroC has all 10 endogenous aromatic amino acids within the C-IDR mutated to alanines. aroNC contains all 19 aromatic amino acid mutations. The aromatic mutant constructs contain the same S33Y mutation as β-catenin*. (B) Representative images from an in vitro droplet formation assay with the indicated mutants. Droplet assays were performed in 300 mM NaCl and 10% PEG-8000. Scale bar = 20 μm. (C) Representative images from an in vitro droplet formation assay testing eGFP-β-catenin* sensitivity to 1,6-hexanediol and 2,5-hexanediol (used as a control); 8 μm eGFP-β-catenin* protein was exposed to the hexanediols prior to PEG-8000 (HD -> PEG) or PEG-8000 prior to the hexanediols (PEG -> HD), and 10% hexanediol and 10% PEG-8000 was used. Scale bar = 20 μm. IDR, intrinsically disordered region.

condensate formation using a buffer containing 10% polyethylene glycol 8000 (PEG-8000) as a crowding agent, following standard experimental guidelines [29]. At relatively low concentrations where β-catenin* formed droplets, the aromatic mutants are deficient in condensate formation (Fig 1B). aroN and aroC displayed a similar defect in droplet formation, while aroNC was more severe. At high concentrations, the aromatic mutants formed condensates at similar levels as β-catenin*. The S33Y point mutation that is present in all 4 mutants does not affect the ability of wild-type β-catenin or aroNC to form condensates (S2 Fig). Consistent with

Zamudio and colleagues, eGFP alone or eGFP-β-catenin with both IDRs deleted (ΔNC; Fig 1A) are incapable of condensate formation (Fig 1B) [20]. Our results demonstrate that both the N-IDR and C-IDR contribute to β-catenin condensation, and these IDRs contain additional sequence information beyond aromatic residues that facilitate droplet formation.

To further examine the properties of the β-catenin* condensates we generated, we made use of the alcohol 1,6-hexanediol (1,6-HD), which is commonly used to inhibit biomolecular condensation [30,31]. The isomer 2,5-hexanediol (2,5-HD) does not disrupt condensation and is used as a control. Droplet formation of β-catenin* was sensitive to 1,6-HD, especially when the alcohol was added prior to the PEG-8000 crowding agent (Fig 1C). When 1,6-HD was added after PEG-8000, some condensation was still observed. The sensitivity to 1,6-HD is consistent with the idea that these assemblies are driven by hydrophobic interactions [31–33], but once the droplets form, some may transition from a phase separated droplet to a more static hydrogel, which can resist the effects of 1,6-HD [34].

In living cells, β-catenin regulates gene expression through interactions with many proteins, the most prominent of which are members of the TCF/LEF family of transcription factors [3]. The central Arm repeats of β-catenin bind to the N-terminus of TCFs, as shown by traditional protein interaction assays and X-ray diffraction of co-crystals [10]. To determine if β-catenin could form heterotypic in vitro condensates with a TCF/LEF member, we expressed mCherry fused to human LEF1 (Figs 2A and S4A for sequence information). IUPred2A and AlphaFold analysis of LEF1 predicts that most of the protein is disordered, except for the DNA-binding HMG domain (S1C and S1D Fig). Purified mCherry-LEF1 can form concentration-dependent droplets in vitro (Fig 2B). Equal amounts of mCherry-LEF1 and eGFP (used as a control for subsequent comparisons) were present in each reaction. Control heterotypic in vitro droplet formation assays show that eGFP does not co-localize with mCherry-LEF1 (S5A Fig) and mCherry does not co-localize with eGFP-β-catenin* (S5B Fig). Next, we performed a dose with equal molar amounts of β-catenin* and LEF1 (Fig 2C). There is a high degree of co-localized fluorescent signal from both eGFP and mCherry, indicating a high degree of miscibility of β-catenin* and LEF1 droplets. Relative to β-catenin*, aroNC exhibits reduced co-localization with LEF1 (Fig 2D), as does ΔNC (Fig 2E). Additionally, eGFP-β-catenin* enhanced the size of mCherry-LEF1 condensates while eGFP-aroNC and eGFP-ΔN did not have a detectable effect (S5C and S5D Fig).

The degree of co-localization in individual BMCs was quantified with line traces, and as expected, the strongest colocalization was with β-catenin* and LEF1, followed by aroNC and ΔNC (Figs 2F–2H and S6). The results indicate that β-catenin is localized to LEF1 condensates by 2 primary interactions. The traditional Arm repeat-LEF1 interaction makes a detectable contribution, based on the ability of LEF1 to recruit ΔNC into a mixed condensate (Fig 2E). However, the presence of the terminal IDRs greatly enhanced the ability of β-catenin and LEF1 to form mixed condensates (Fig 2C) and the aromatic residues in these IDRs are required (Fig 2D).

## Aromatic amino acid residues within β-catenin's terminal IDRs contribute to homotypic and heterotypic condensate formation in cultured cells

We wanted to expand on the in vitro BMC results by testing whether the aromatic residues within β-catenin's IDRs are important for BMC formation in living cells. To do this, we overexpressed eGFP-β-catenin* and eGFP-tagged aromatic mutants in HEK293T β-catenin KO cells. eGFP-β-catenin* forms spherical puncta in the nucleus that are consistent with BMCs, while the aromatic mutants do not (Fig 3A and 3B). The difference between the ability of eGFP-β-catenin* and the aromatic mutants to form BMCs in cells is starker than in vitro, as

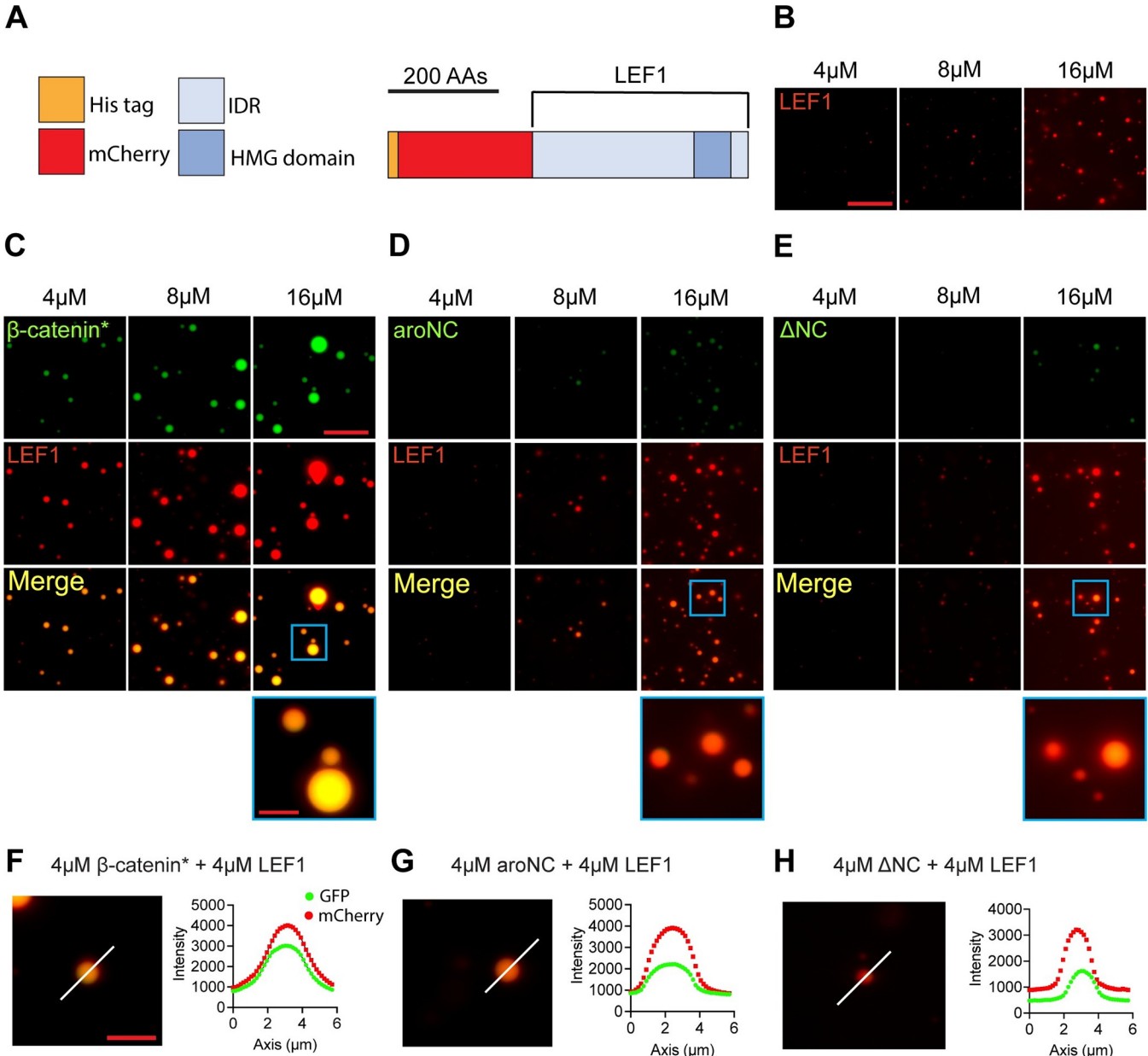

**Fig 2. The β-catenin terminal IDRs promote β-catenin incorporation into LEF1 condensates in vitro.** (A) Cartoon representation of the mCherry-LEF1 construct. (B) Representative images from an mCherry-LEF1 in vitro droplet formation assay. Equal amounts of mCherry-LEF1 and eGFP (used as a control) were present in each reaction. Assays were performed in 300 mM NaCl and 10% PEG-8000. Scale bar = 20 μm. (C–E) Representative images from heterotypic in vitro droplet formation assays. (C) Equal amounts of eGFP-β-catenin* and mCherry-LEF1, (D) eGFP-aroNC and mCherry-LEF1, and (E) eGFP- ΔNC and mCherry-LEF1 were mixed, resulting in total protein concentrations of 4 μm, 8 μm, and 16 μm. Reaction conditions are the same as panel B. Scale bar = 20 μm, inset scale bar = 5 μm. (F–H) Line plots showing fluorescent intensity across a droplet. (F) eGFP-β-catenin* + mCherry-LEF1, (G) eGFP-aroNC and mCherry-LEF1, and (H) eGFP-ΔNC and mCherry-LEF1. Increased fluorescent signal for both proteins across a line indicates co-localization and the white lines represent the plotted trace. Scale bar = 5 μm. Summary data displayed in Fig 2 can be found in S1 Data. IDR, intrinsically disordered region.

the aromatic mutants have seemingly no residual capacity to form detectable BMCs. Approximately 40% of the cells expressing eGFP-β-catenin* display puncta, while the remainder have a diffuse signal in both the cytosol and the nucleus, similar to that observed with the aromatic mutants (S7 Fig).

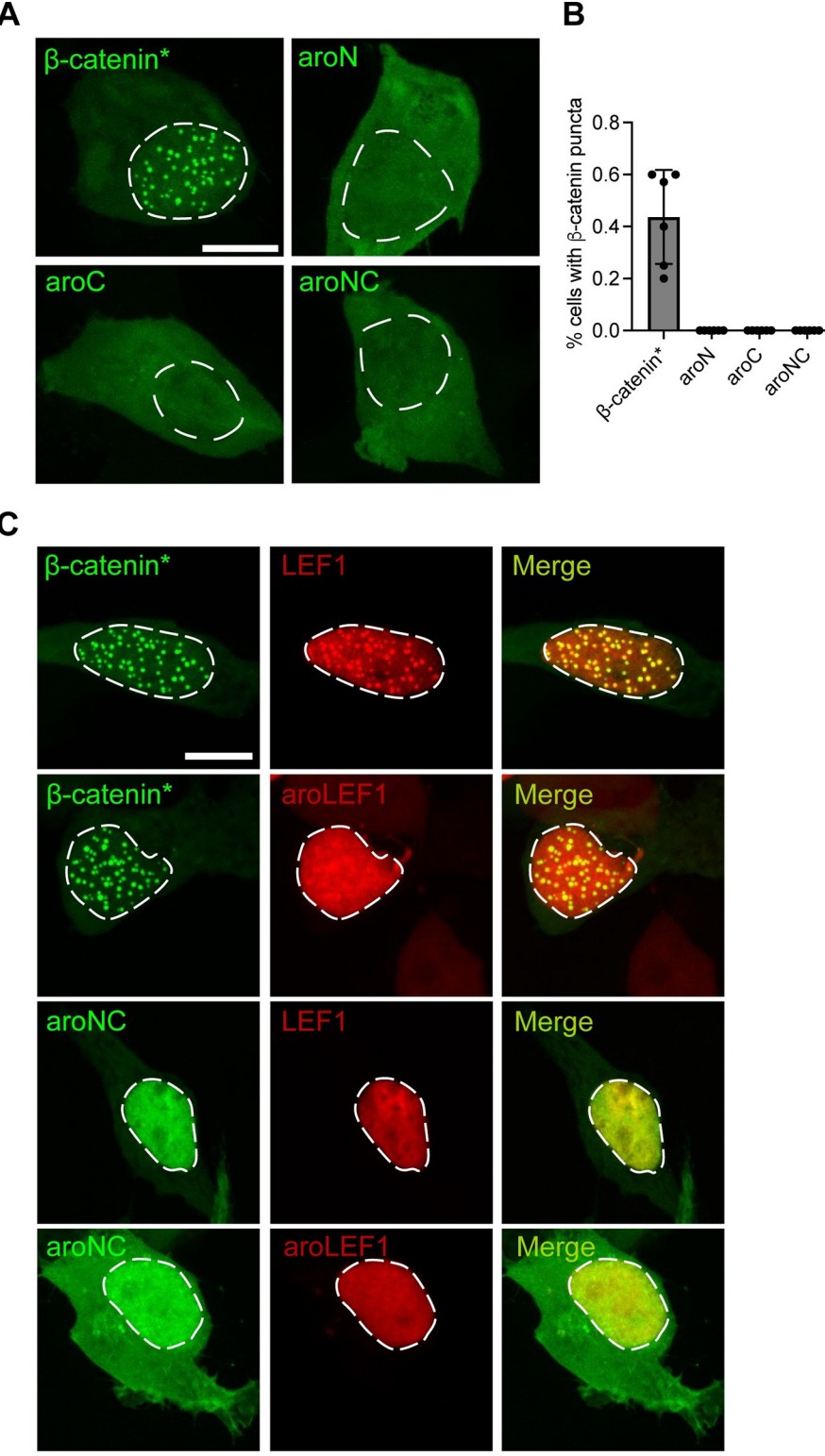

**Fig 3. Aromatic amino acid residues within β-catenin and LEF1 promote biomolecular condensate formation in vivo.** (A) Representative confocal images of HEK293T β-catenin KO cells expressing β-catenin mutants. Cells were transiently transfected with constructs encoding the FLAG-eGFP-β-catenin mutants. Scale bar = 10 μm. (B) Quantification of the percentage of cells that exhibited puncta. Data are presented as mean ± SD (*n* = 3). (C) Representative confocal images of HEK293T β-catenin KO cells expressing the indicated β-catenin and LEF1

constructs. Cells were transiently transfected with constructs encoding the FLAG-eGFP-β-catenin mutants and the FLAG-mCherry-LEF1 mutants. Scale bar = 10 μm. Summary data displayed in Fig 3 can be found in S1 Data.

Recent work has shown that LEF1 can form heterotypic BMCs with β-catenin in vivo, and IDR deletion experiments show that LEF1's N-terminal IDR is important for BMC formation and transcriptional activity [22]. We also found that eGFP-β-catenin* co-localized with mCherry-LEF1 displaying heterotypic puncta in HEK293T β-catenin KO cells (Fig 3C). To test whether the aromatic residues with LEF1's N-terminal IDR are required for co-localization with β-catenin, we generated aroLEF1, where 14 aromatic residues were substituted to alanines (S4A Fig). mCherry-aroLEF1 had no effect on eGFP-β-catenin* puncta formation, but did not accumulate these puncta, though it still accumulated in nuclei (Figs 3C and S8). Expressing eGFP-aroNC with either mCherry-LEF1 or mCherry-aroLEF1 did not lead to puncta formation by either protein, but it did increase the nuclear localization of eGFP-aroNC (Fig 3C). While previous studies have demonstrated the importance of TCF's N-IDR for BMC formation and activity [22,35], our work shows that aromatic amino acid residues within the IDR are essential drivers of condensation.

## Aromatic amino acid residues within β-catenin's IDRs are critical for nuclear function in cultured human cells

To test whether the ability of β-catenin to form homotypic and heterotypic condensates in vitro is relevant to its ability to activate Wnt targets in cultured cells, we expressed β-catenin*, aroN, aroC, and aroNC in HEK293T cells in the presence of several Wnt reporters. Using either the synthetic reporter TopFlash (containing 6 copies of high affinity TCF binding sites) [36] or a reporter with an endogenous WRE from the *Axin2* locus, known as CREAX [37]. We found that all 3 β-catenin aromatic mutants had greatly reduced transcriptional activation activity (Fig 4A) even though their expression levels were slightly higher than β-catenin* (Fig 4B). The defect in reporter gene activation was also observed when wild-type β-catenin and an aroNC mutant with serine at position 33 were tested (S2B Fig).

The reduced transcriptional activity of the aromatic mutants can be potentially explained by a reduced ability to translocate to the nucleus relative to wild-type β-catenin. To address this issue, immunofluorescence (IF) was performed using an anti-FLAG antibody. We observed no detectable difference in the ability of these proteins to accumulate in the nucleus at similar levels as β-catenin* (Fig 4C and 4D). These observations were corroborated by cell fractionation that indicated no deficit of aroNC protein in the nuclear fraction compared to β-catenin*, as judged by western blots (Fig 4E and 4F). These results suggest that mutating the aromatic residues in β-catenin's terminal IDRs does not have a detectable effect on the protein's ability to translocate to the nucleus. While some nuclear puncta were observed in cells expressing FLAG-β-catenin*, they were smaller in size, fewer in number, and only observed under high magnification (S9 Fig).

The alteration of so many aromatic residues in the terminal IDRs of β-catenin raises the concern that the defect in signaling activity could be due to disruption of protein–protein interactions with known binding partners. To address this, we performed co-IP experiments with FLAG-tagged β-catenin*, aroN and aroNC, probing for interactions with TCF7, TCF7L2, BRG1, p300, and CBP. These proteins are required for Wnt/β-catenin signaling and physically interact with β-catenin [5,38,39]. We found these proteins were pulled down to similar degrees by all 3 β-catenin proteins (S10 Fig). These experiments indicate that the aroN and aroNC mutations do not have a global effect on TCF and co-activator binding.

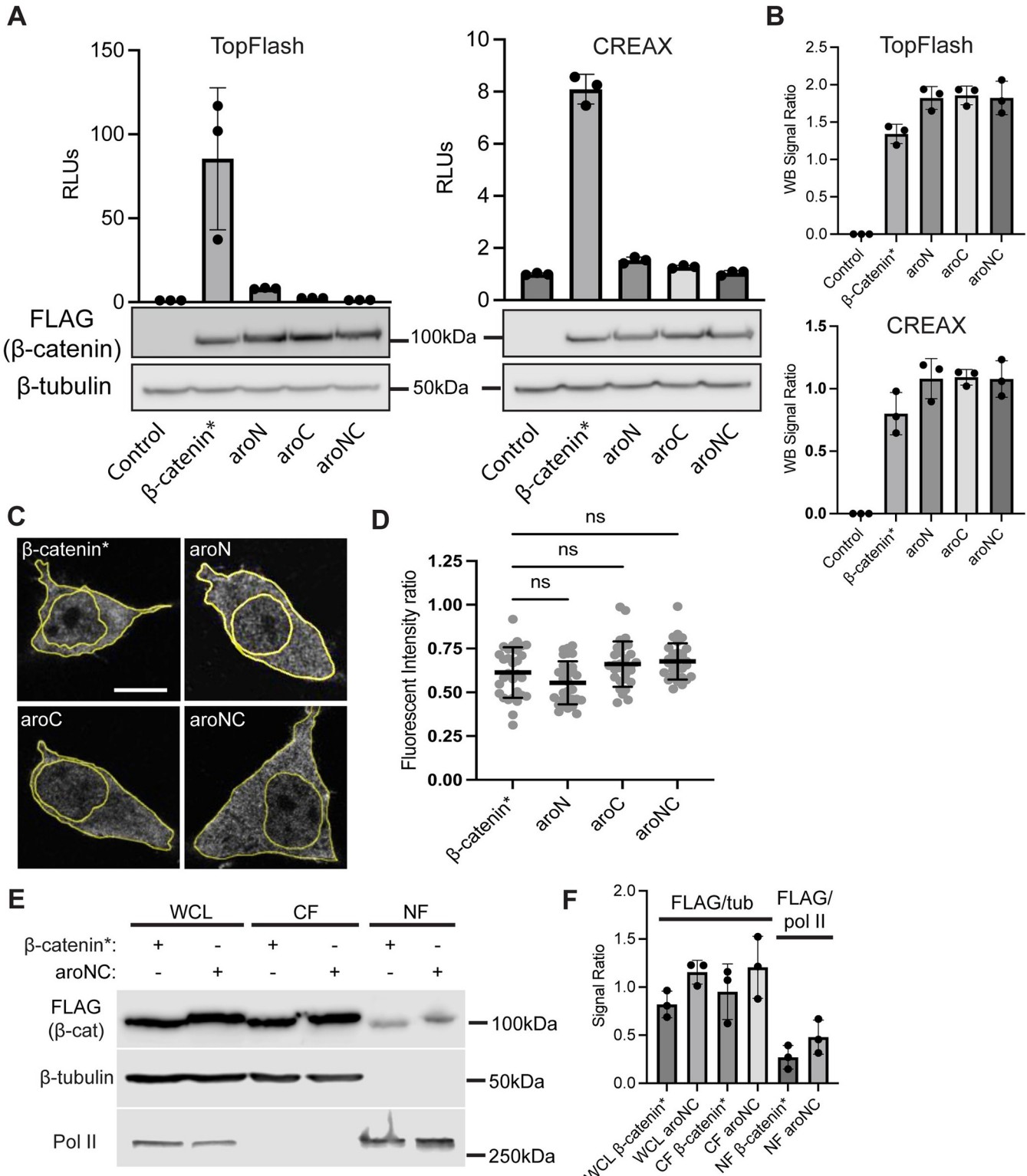

**Fig 4. Aromatic residues within β-catenin's terminal IDRs are required for reporter gene activation and not nuclear accumulation.** (A) TopFlash (left) or CREAX (right) luciferase reporter activity induced by β-catenin* or the aromatic mutant constructs in HEK293T cells. Cells were transfected with separate plasmids encoding the reporter genes and the FLAG-β-catenin mutant constructs or pcDNA3.1 as a negative control. Corresponding western blots show relative expression of the FLAG-β-catenin mutant constructs. (B) Quantification of western blots from 3 independent replicate experiments for A. Data presented as the ratio of FLAG signal intensity to β-tubulin signal intensity. (C) Representative IF images of HEK293T cells for the indicated FLAG-β-catenin

mutants. Cells were transfected with plasmids encoding the FLAG-β-catenin mutant constructs. IF was performed 24 h after transfection and nuclei were stained with DAPI. The borders of the cell and nucleus are highlighted. Scale bar = 10 μm. (D) Quantification of IF showing no significant difference in nuclear localization between β-catenin* and aromatic mutants. Data is presented as a ratio of the fluorescent intensity within the nucleus to the fluorescent intensity outside the nucleus. (E) Western blot showing nuclear fractionation samples from HEK293T cells expressing FLAG-β-catenin* or FLAG-aroNC. (F) Quantification of western blots from 3 independent experiments. Data presented as the ratio of FLAG signal intensity to either β-tubulin or RNA pol II signal intensity. All data are presented as mean ± SD; *p*-values were calculated by one-way ANOVA followed by Dunnett's test. ns = *p* > 0.05. Summary data displayed in Fig 4 can be found in S1 Data. IDR, intrinsically disordered region; IF, immunofluorescence.

Overall, these results demonstrate the importance of the IDR aromatic residues for the ability of nuclear β-catenin to activate Wnt target gene expression. In addition, the defect in aroN reveals a previously unappreciated role for the N-terminus of β-catenin in transcriptional activation, which we suggest is due to the defect in the ability of the aromatic mutants to efficiently form BMCs.

Our studies clearly indicate that aromatic residues in both IDRs are necessary for signaling activity, but further mutagenesis is needed to determine whether all aromatic residues equally contribute to β-catenin activity. One popular model of condensation driven by aromatic residues posits that aromatic residues flanked by polar amino acids (i.e., stickers) are the drivers of IDR-IDR interactions, while other aromatics serve as "spacers" [40]. To test this sticker/spacer model in the context of β-catenin transcriptional activity, we constructed 4 additional mutants (S11A Fig for sequence information). Per the model, we mutated 5 potential stickers and 4 potential spacers in N-IDR, and 5 potential stickers and spacers each in the C-IDR. These mutants were tested for activity using TopFlash, CREAX, and Defa5 (a reporter with an endogenous WRE from the Defa5 locus [41]) reporter assays. All 4 mutants displayed reduced activity but were more active than their aroN or aroC counterparts (S11B and S11C Fig). While N-sticker mutations led to greater reductions in activity than the N-spacer mutations, the putative spacer aromatics in the C-IDR were more critical for activity than the putative stickers. This inconsistency between IDRs does not support a strict sticker/spacer model for β-catenin's IDRs; the results are more consistent with a model where many/most of the aromatic residues in the IDRs contribute to biological activity.

Tyrosine phosphorylation plays an important role in β-catenin regulation [42]. For example, the phosphorylation of 2 residues, tyrosines 142 and 654, promotes the dissociation of β-catenin from adherens junctions and increases Wnt signaling [42–44]. Additional tyrosine residues are phosphorylated throughout β-catenin, such as tyrosines 331 and 333 which may have a similar activating effect [45]. Since our aromatic β-catenin mutants contain multiple tyrosine mutations across both IDRs, we wanted to ensure that these mutations were not affecting β-catenin transcriptional activity. We generated a Y-to-F β-catenin mutant in which all the tyrosine residues that were mutated in the aroNC construct were changed to phenylalanine (S12A Fig). This Y-to-F mutant activates the TopFlash reporter as well as β-catenin*, indicating transcriptional activity is not compromised (S12B Fig). co-IP analysis shows that the overall level of tyrosine phosphorylation on the aroNC mutant was also not detectably different than β-catenin* (S12C Fig). These data suggest that the aroNC mutation does not have a large effect on tyrosine phosphorylation, and the observed differences in activity are not due to mutating the tyrosine residues within β-catenin's terminal IDRs.

To extend our analysis beyond reporter genes, we examined the role of the IDR aromatic residues in β-catenin's ability to regulate endogenous Wnt targets. We generated stable HeLa cell lines which expressed β-catenin* and the aromatic mutants from a DOX-inducible expression cassette via lentiviral transduction. We chose to assay the Wnt target genes *Axin2*, *Sp5*, and *Notum* as they are strongly activated by Wnt signaling in HeLa cells [46,47]. qPCR analysis of HeLa cells expressing β-catenin* and the aromatic mutants indicates that the relationship between aromatic amino acid residues and gene regulation is more complex than the reporter

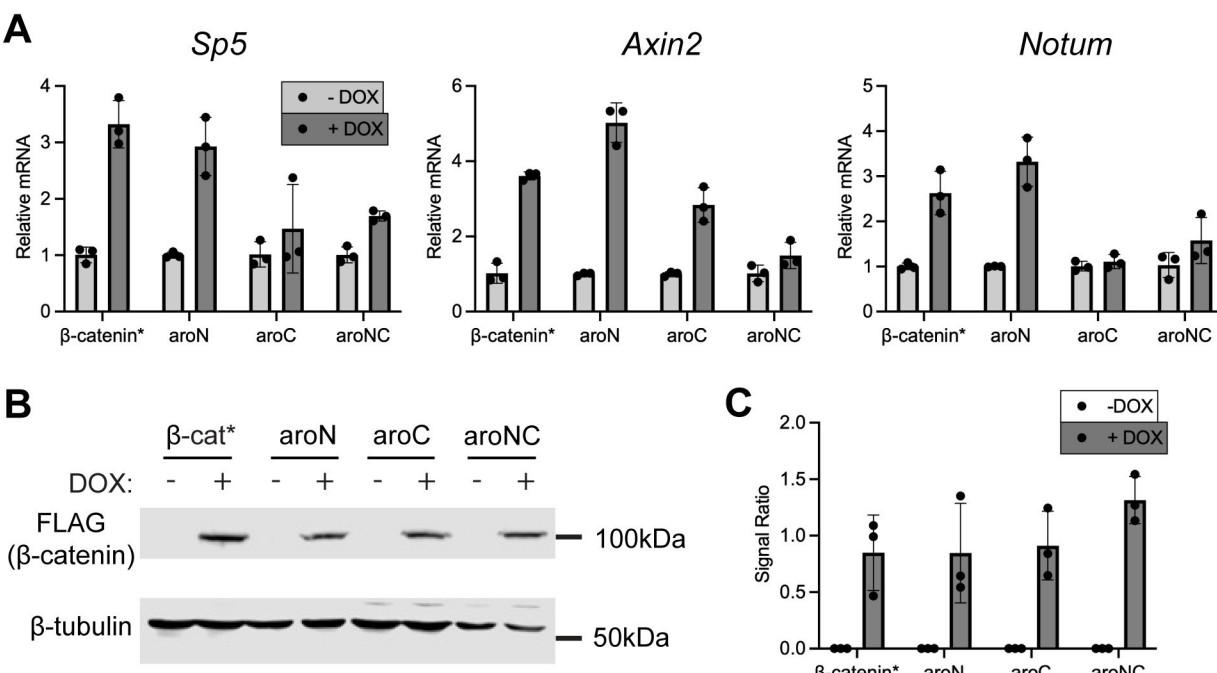

**Fig 5. Select Wnt target genes exhibit different sensitivities to β-catenin aromatic mutant constructs.** (A) qRT-PCR analysis of 3 Wnt target genes (*Sp5*, *Axin2*, and *Notum*) in HeLa cells that were stably transformed with DOX-inducible, β-catenin mutant expression vectors. Cells were treated with DOX for 24 h prior to harvesting. (B) Western blot analysis of DOX-treated HeLa cell lysate. Lysate samples correspond to the qPCR data. α-FLAG blot shows β-catenin expression. α-tubulin was used as a loading control. (C) Quantification of western blots from 3 independent experiments. Data presented as the ratio of FLAG signal intensity to tubulin. All data presented as mean ± SD (*n* = 3). Summary data displayed in Fig 5 can be found in S1 Data.

activity (Fig 5A). As expected, β-catenin* activates all the tested Wnt target genes. Interestingly, aroN also activated all the tested Wnt targets, analogously to β-catenin*. aroC and aroNC were transcriptionally deficient relative to β-catenin*. aroC was able to activate *Axin2*, but not *Sp5* or *Notum* and aroNC was able to weakly activate *Sp5* but not *Axin2* or *Notum*. The different constructs were expressed at approximately the same level (Fig 5B and 5C). The nuclear localization of aroNC relative to β-catenin* was not deficient in HeLa cells (S13A and S13B Fig). Similar to HEK293T cells, aroNC was also deficient in activating TopFlash in HeLa cells (S13C Fig).

This gene expression data is different from the luciferase reporter data as some β-catenin aromatic mutants still have considerable capacity to activate endogenous genes. The observation that the aromatic mutants have a different effect on genes within the same cell type indicates that there can be gene-specific requirements for BMCs in Wnt target gene regulation. It is also noteworthy that while the activity of the aromatic mutants can be ordered (aroN>aroC>aroNC), all 3 are deficient in forming puncta when overexpressed in cells (Fig 3A). This could indicate that some β-catenin–dependent gene activation is BMC independent, but it may also reflect the limitations of the eGFP-β-catenin puncta assay, i.e., only the largest BMCs are evident with the visualization employed.

## β-catenin/Armadillo IDR aromatic residues are critical for function in *Drosophila* development

To test whether the importance of aromatic residues for β-catenin function is conserved across species, we examined their role in the activity of the fly β-catenin, Armadillo (Arm). Wingless

(Wg)/Arm signaling is required throughout *Drosophila* development and has been intensively studied in *Drosophila* embryos and larval imaginal discs [48,49]. Arm, like human β-catenin, contains 9 and 10 aromatic residues in its N-IDR and C-IDR, respectively. Seven of the N-terminal aromatics and 5 of the C-terminal aromatics are conserved across multiple species. We constructed 5 Arm transgenes, under the control of the Gal4-UAS expression system (S14 Fig for protein sequences). Like the β-catenin mutants that we constructed, aroN has all 9 N-IDR aromatic residues mutated to alanine, aroC has all 10 C-IDR aromatic residues mutated to alanine, and aroNC has both sets of mutations. aroN-cons has the 7 conserved aromatic residues mutated, with Y3 and Y17 remaining. aroC-cons has the 5 conserved aromatic residues mutated, with 723, F743, Y748, Y803, and F819 remaining. All the aromatic mutant Arm proteins have the stabilizing T52A and S56A mutations, which are also found in Arm*. These transgenes were integrated into 2 locations in the fly genome using phiC31 landing sites [50], ensuring similar levels of transcription. These UAS-transgenes were expressed in various tissues and their effect on Wg/Arm readouts were assayed.

It has been previously shown that expressing Wg agonists in the larval eye via the GMR-Gal4 driver results in smaller eyes due to increased apoptosis [51,52]. Using GMR-Gal4 to overexpress UAS-Arm*, the constitutively active mutant, we observed a reduction in adult eye size, and the loss of pigmentation and cone cells. The aromatic Arm mutants (which contain the stabilizing mutants of Arm*) exhibit a small reduction in eye size, while maintaining a normal eye morphology (Fig 6A and 6C). Cut immunostaining of pupal eye tissue shows that overexpression of Arm* in the larval eye disrupted both the organization of ommatidia and the regular arrangement of Cut+ cone cells within each ommatidium. Overexpression of the aromatic Arm mutants does not affect Cut+ cone cell patterning within the tissue (Fig 6B). These mutants are expressed at similar levels, as detected by IF (Fig 6D). These data indicate that the aromatic residues within both IDRs are essential for Arm function in the developing *Drosophila* eye, as mutating them strongly abrogates Arm's signaling activity in this tissue.

Wg/Arm signaling is important for patterning the wing imaginal disc during larval development. In this tissue, Wg is expressed across the dorsal/ventral boundary in a stripe, regulating targets at short and long ranges from the source of Wg synthesis [48]. This gradient of signaling can be detected with a synthetic Wg reporter containing 4 Grainy head binding sites upstream of 4 HMG-Helper site pairs, arranged for high affinity binding by Pangolin (the fly TCF ortholog) [53–55]. Decapentaplegic-Gal4 (DPP-Gal4) was used to overexpress Arm* and the aromatic mutants in a stripe pattern that is perpendicular to the endogenous Wg expression stripe (Fig 7A, white arrowhead). To prevent major disruption to the wing disc morphology, the Gal80$^{ts}$ system was used to inhibit Gal4 activity (and Arm protein expression) until 18 h prior to fixation. Arm* overexpression resulted in the strongest ectopic activation of the Wg reporter, the aroN and aroN-cons constructs exhibited a moderate activation of the reporter, and the aroC, aroC-cons, and aroNC mutants exhibited the weakest ectopic activation (Fig 7A, top). The observations and categorizing the constructs into strong/moderate/weak activators is supported by quantification of the fluorescent reporter activity and subsequent statistical analyses (Fig 7B). All mutant constructs were expressed at similar levels, as detected by FLAG IF (Fig 7A, bottom). These reporter data show that all the aromatic mutants have some residual capacity to activate transcription. These data are distinct from the developing eye, in which the aromatic mutants had little residual activity. This difference in sensitivity to loss of aromatic residues could be due to differences in the degree for BMC-dependency for activation of Wnt targets in different tissues.

Wg/Arm signaling also plays a key role in patterning the *Drosophila* embryo. Segments of the ventral embryonic epidermis feature a characteristic, trapezoidal-shaped belt of denticles. These denticle belts are separated by regions of naked cuticle. The establishment of denticle

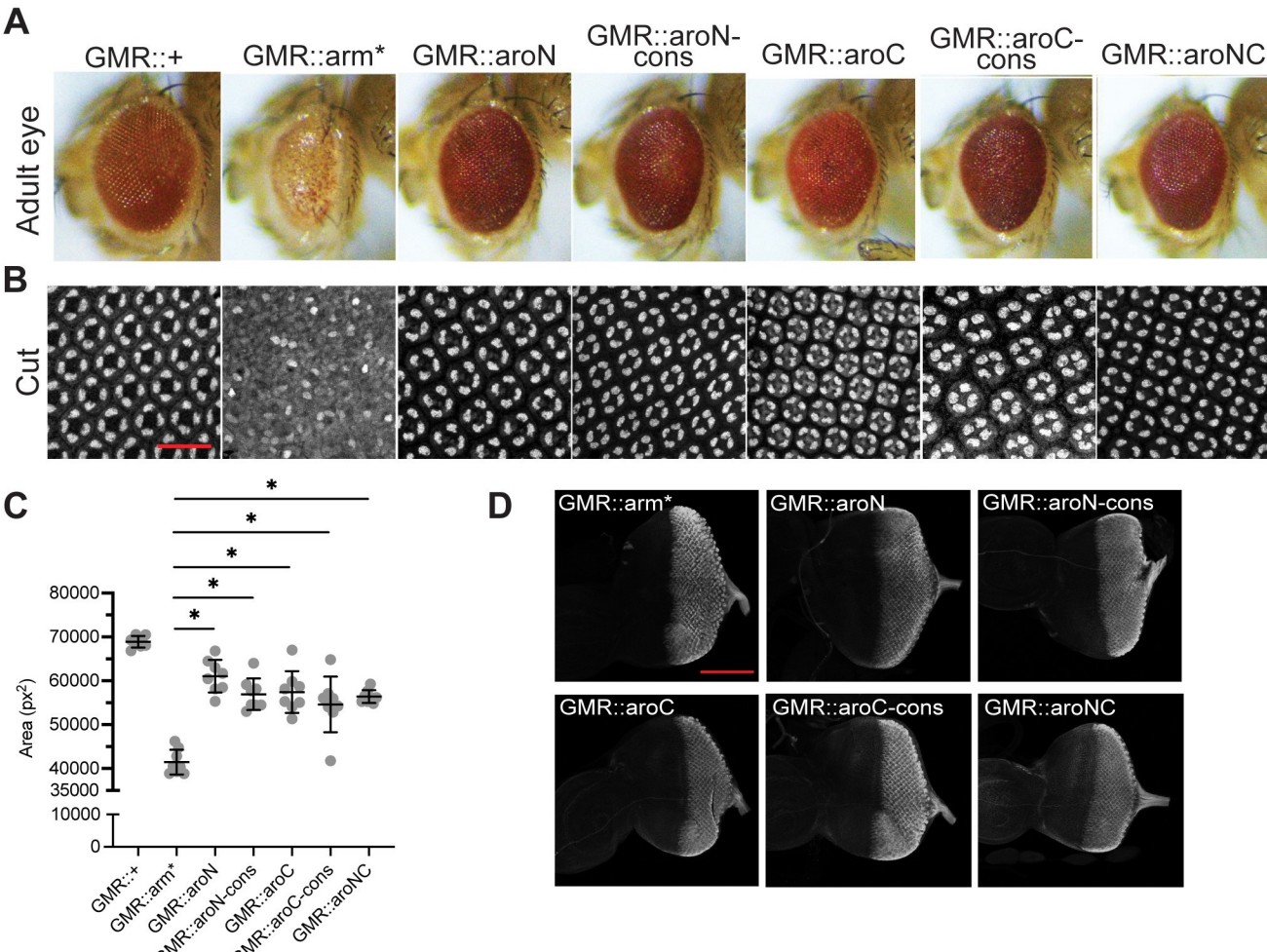

**Fig 6. β-catenin/Arm activity in the adult *Drosophila* eye is attenuated by aromatic amino acid mutations within the terminal IDRs.** (A) Micrographs of adult *Drosophila* eyes containing P[GMR-Gal4] and various P[UAS-Arm] transgenes. (B) Representative images of pupal *Drosophila* eye tissue immunostained for the cone cell marker Cut. Stabilized Arm (Arm*) disrupts cone cell specification while aromatic mutants do not. Scale bar = 20 μm. (C) Quantification of adult *Drosophila* eye area. Data are presented as mean ± SD ($n = 8$); $p$-values were calculated by one-way ANOVA followed by Dunnett's test. * = $p < 0.05$. (D) Expression of the various Arm constructs is constant during late larval eye development. Representative images of larval *Drosophila* eye antennal discs immunostained for FLAG, representing β-catenin mutant expression. Scale bar = 100 μm. Summary data displayed in Fig 6 can be found in S1 Data. IDR, intrinsically disordered region.

belts and naked cuticle is regulated by Wnt signaling [56]. Increasing Wnt signaling throughout the embryo expands regions of naked cuticle at the expense of denticle band formation, and conversely, loss of Wnt signaling leads to ectopic denticle formation and a failure to form naked cuticle [57]. To test the effect that our aromatic mutants have on regulating this phenotype, we overexpressed our constructs to similar levels using a stock containing 2 constitutive Gal4 drivers, Daughterless-Gal4 (Da-Gal4) and Arm-Gal4, both of which are active throughout the embryonic epidermis (S15 Fig) [58,59]. When crossed to this Gal4 driver stock, UAS-Arm* displays a classic naked cuticle phenotype, with a 100% phenotype penetrance (Fig 7C). In contrast, overexpression of the aromatic mutants all resulted in a similar phenotype: a partial loss of denticle formation along the ventral midline (Fig 7C). These phenotypes were highly penetrant and consistent with a moderate level of Arm signaling activity. Our data indicates that the loss of aromatic residues in N-IDR and as little as 5 aromatic residues within

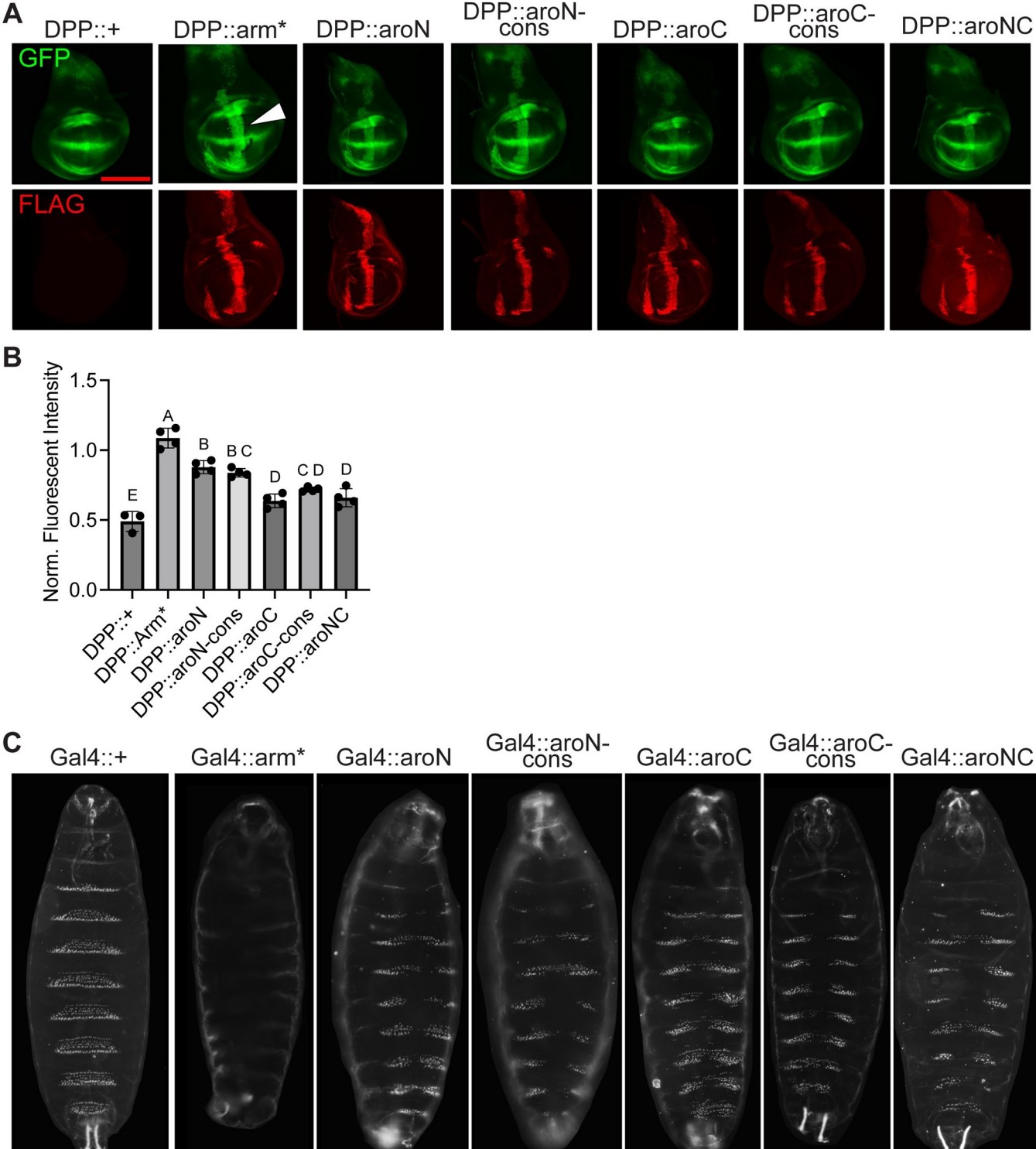

**Fig 7. Aromatic β-catenin/Arm mutants exhibit different levels of activity in wing imaginal discs and embryonic epidermis.** (A) Representative images of late third instar wing imaginal discs showing expression of a synthetic Wnt GFP reporter combined with a P[Dpp-Gal4] driving expression of various P [UAS-Arm] transgenes. Discs were also immunostained with α-FLAG to detect expression of the various Arm mutants. Gal4 activity was restricted to 18 h before fixation using a Gal80$^{ts}$ transgene. Scale bar = 100 μm. (B) Quantification of the synthetic Wnt GFP reporter activity. Reporter activity driven by ectopic expression of Arm mutants was normalized to reporter activity driven by endogenously expressed Wg/Arm signaling. Letters above bars indicate statistical significance ($p < 0.05$, calculated by one-way ANOVA followed by Dunnett's test). Bars with the same letter above them are not significantly different, bars

with different letters are. (C) Representative darkfield images showing the ventral side of late embryonic *Drosophila* cuticles containing P[Da-Gal4], P[Arm-Gal4], and 2 copies of the various P[UAS-Arm] transgenes. Scale bar = 100 μm. Summary data displayed in Fig 7 can be found in S1 Data.

the C-IDR (i.e., aroC-cons) compromise Arm's signaling activity to similar extents. The results also indicate that in the embryo, as in the wing disc, the aromatic Arm mutants retain an appreciable degree of activity.

To test the ability of Arm aromatic mutants to rescue an *arm* loss-of-function phenotype, we expressed our transgenes at a reduced level (Da-Gal4 plus 1 copy of a UAS-Arm transgene). We reasoned that at this lower level of expression, the transgenic Arm* would be able to rescue the severe cuticular phenotype of embryos lacking zygotic *arm* gene activity. In embryos lacking zygotic *arm* activity, the size of the embryo is greatly reduced, the head and posterior structures are missing or malformed, and the "naked" cuticle normally found on the posterior portion of each segment is absent, instead displaying ectopic denticles. Indeed, expression of Arm* was able to rescue the *arm* mutant phenotype to a high degree with 100% penetrance (Fig 8A–8C). Expression of the aromatic mutants also resulted in significant rescue (Fig 8D–8H): the overall size of these embryos is similar to wild-type embryos and the Arm* rescue, head structures are largely restored and there is significant recovery of the posterior-most structures. However, the degree of rescue was significantly less for the aromatic mutants compared to Arm*, as evidenced by the presence of excess denticles in all the abdominal segments. The difference in rescue may be reflected in the ability of these overexpression backgrounds to drive a naked cuticle phenotype in a wild-type Arm genetic background. Single-copy overexpression of Arm* causes a moderate naked cuticle phenotype, as some denticles are still present (S16A Fig) and aroNC overexpression exhibits weaker activity, as it only disrupts denticle formation near the ventral midline (S16B Fig).

The extent to which aroNC can rescue an *arm* loss of function phenotype relative to Arm* suggests a surprising degree of residual activity. Whether this is related to the residual ability of aroNC to form BMCs or because many molecular targets in embryonic epidermis do not require β-catenin condensation will require additional experiments (see Discussion for further comment).

## Heterologous IDRs can rescue β-catenin signaling activity of an N-IDR deletion mutant

To this point, our mutagenesis approach has correlated a loss of aromatic residues within β-catenin's IDRs with a loss of BMC formation and transcriptional regulation function, providing a link between the ability to form BMCs in vitro and in vivo protein function. This argument is problematic for the C-IDR, which has been implicated in binding to transcriptional co-activators [39]. This caveat is mitigated by the fact that there are no known co-activator binding partners for N-IDR. The deletion of the N-IDR dramatically affected the ability of β-catenin to form droplets in vitro and activate some Wnt targets (Figs 9A, 9B and 10A). This provided an opportunity to test whether these activities could be rescued by adding heterologous IDRs to β-catenin lacking the N-IDR. A collection of IDRs [60] was screened with the following criteria: (1) the IDR must come from a protein with no known nuclear function; (2) must be of similar size as N-IDR (~140aa); and (3) must have a similar frequency of aromatic amino acids. Two IDRs, from human Septin 4 (Sept4) and Sorting nexin 18 (SNX18), met these criteria and were utilized (S17 Fig for protein sequences).

We generated eGFP-tagged β-catenin mutant constructs with the heterologous IDRs at the N-terminus (Fig 9A). We then performed a concentration series in vitro droplet formation

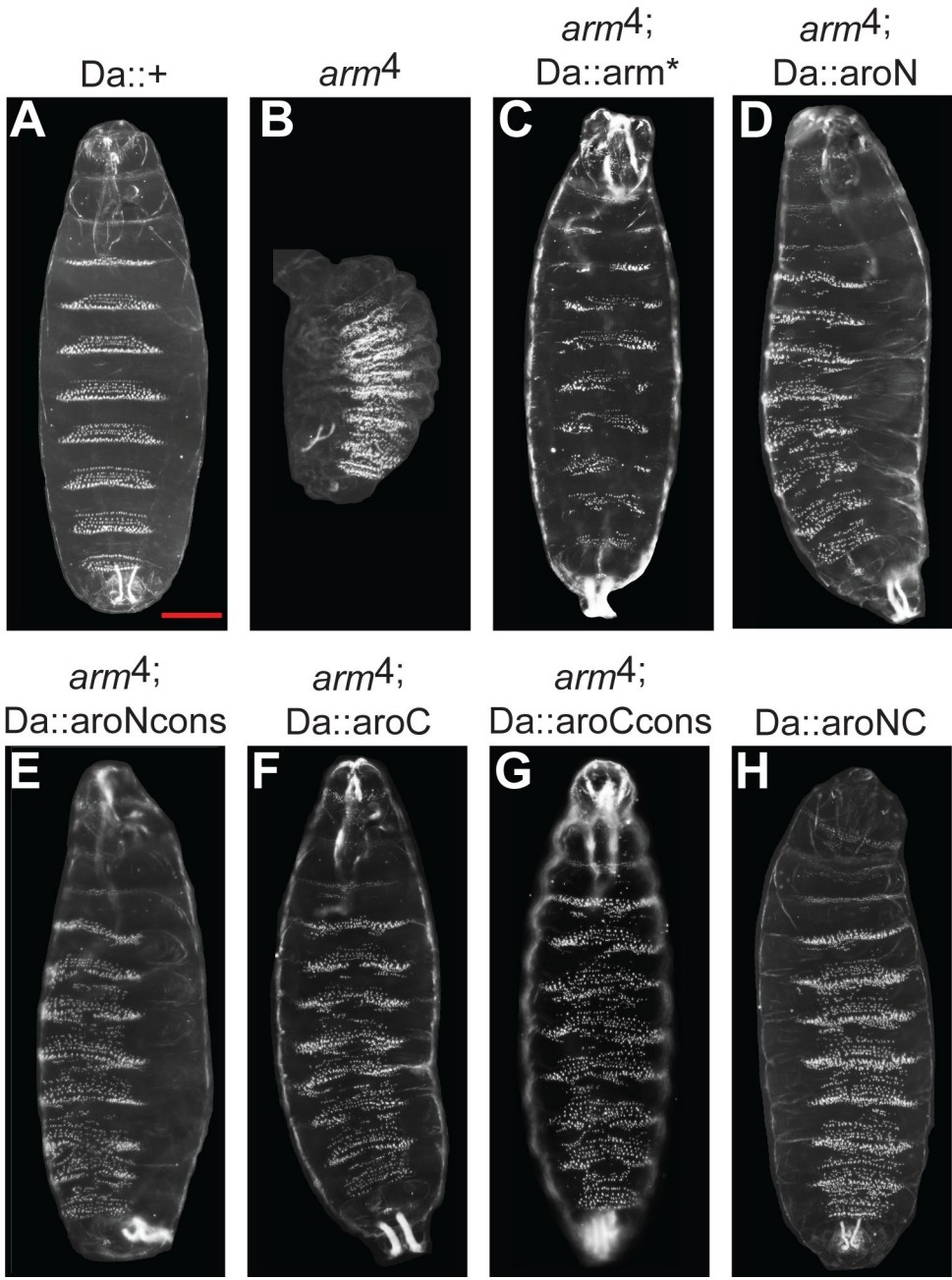

**Fig 8. Aromatic β-catenin/Arm mutants partially rescues an *arm* loss-of-function allele.** (A) Ventral side of a cuticle from a *Drosophila* embryo containing the P[Da-Gal4] transgene. Phenotype is indistinguishable from wild-type. (B) Cuticle of an amorphic *arm* mutant (*arm*4), displaying a classic Wg loss-of-function phenotype. (C) Cuticle of an *arm* mutant embryo containing P[Da-Gal4] and P[UAS-Arm*]. This combination results in nearly complete rescue of the size, head and posterior and lawn of denticle *arm* phenotype with 100% penetrance. (D–H), Cuticles of *arm* mutants embryos containing P[Da-Gal4] and either P[UAS-aroN] (D), P[UAS-aroNcons] (E), P[UAS-aroC] (F), P[UAS-aroCcons] (G), or P[UAS-aroNC] (H). These embryos have rescued the size, head and posterior phenotypes, but not the ectopic denticles.

assay (Fig 9B). Consistent with our previous observations, eGFP-β-catenin* will form spherical BMCs across the concentration range. In contrast, eGFP-ΔN forms fibril-like structures at relatively high concentrations. These fibril-like structures are morphologically distinct from the

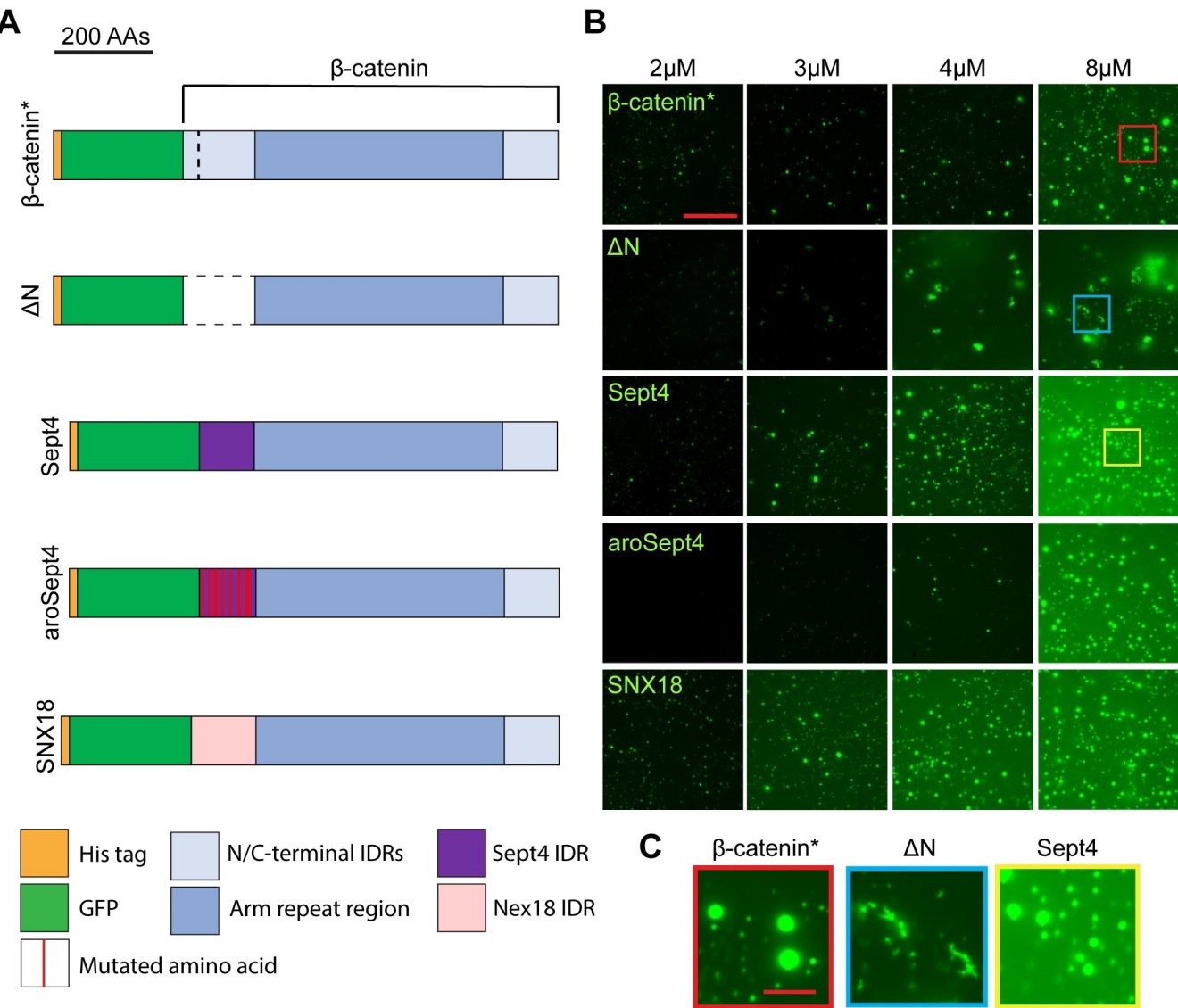

**Fig 9. Heterologous IDRs rescue the in vitro droplet formation of an N-terminal β-catenin deletion mutant.** (A) Cartoon representation of the eGFP-β-catenin* protein, ΔN (amino acids 1–151 deleted), Sept4 (119 amino acids from the Septin4 IDR), aroSept4 (Sept4 IDR with 11 aromatic amino acid mutations), and SNX18 (139 amino acids from the SNX18 IDR). (B) Representative images from a concentration series in vitro droplet formation assay with the indicated mutants. Droplet assays were performed in 300 mM NaCl and 10% PEG-8000. Scale bar = 20 μm. (C) Insets from the highlighted regions of panel B. IDR, intrinsically disordered region.

BMCs formed by the other mutants (Fig 9C). eGFP-Sept4, eGFP-aroSept4, and eGFP-SNX18 form BMCs that are similar in shape to β-catenin, rescuing the fibril-like structures of eGFP-ΔN. Furthermore, mutating the aromatic residues within the Sept4 IDR compromises the ability to form BMCs, reminiscent of the aroN mutant. These data suggest that an N-terminal IDR with sufficient aromatic content is required for efficient BMC formation in vitro and that the primary sequence of β-catenin's endogenous N-IDR is not the driver of BMC formation.

We next wanted to test if these heterologous IDR β-catenin mutants could rescue transcriptional activity of β-catenin. To do this, we utilized the TopFlash luciferase transcriptional reporter in a HEK293T β-catenin KO cell line [61]. We transiently transfected these cells with the reporter and expression constructs for the heterologous IDR β-catenin mutants (Fig 10A).

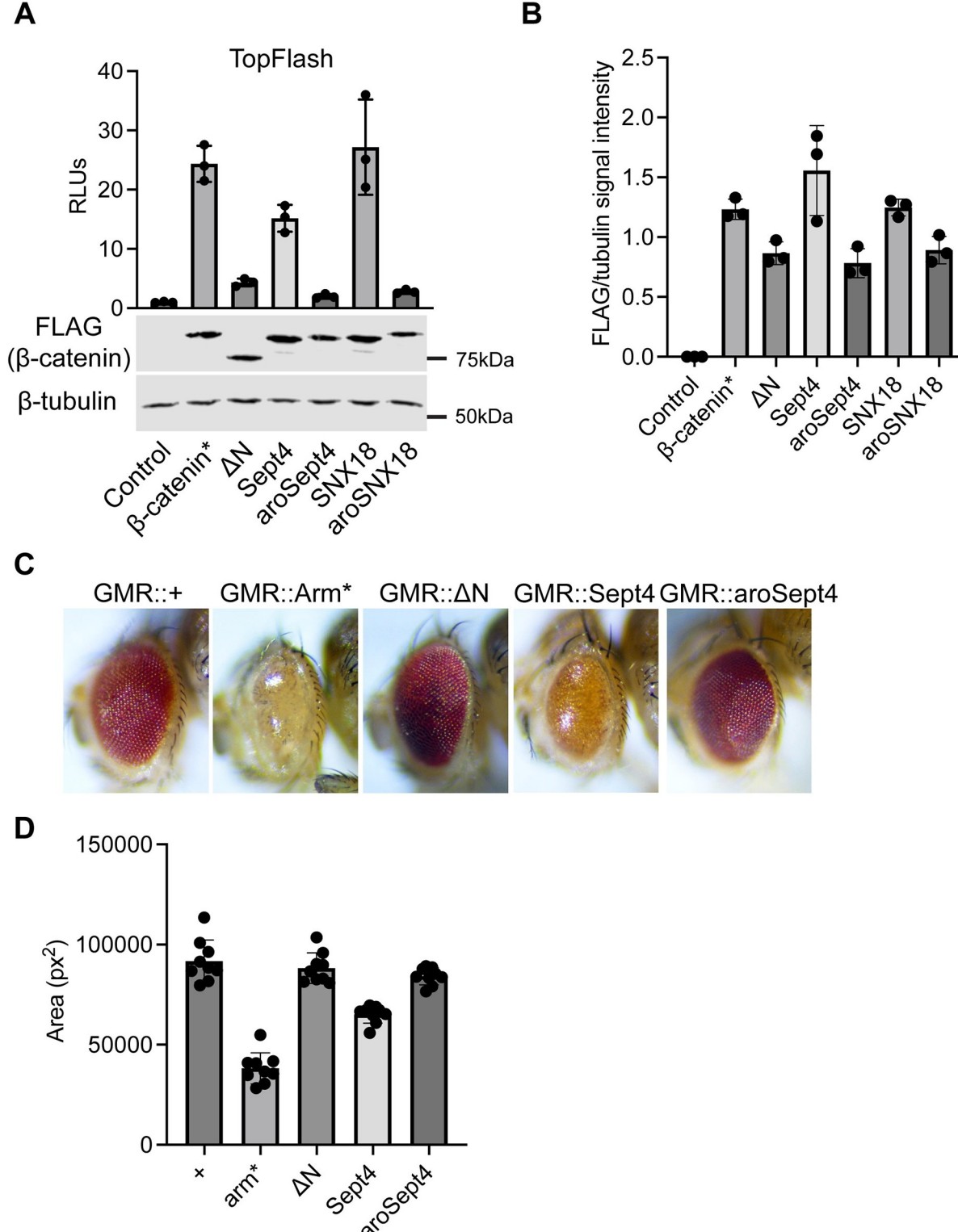

**Fig 10. Heterologous IDRs can rescue the activity of an N-terminal β-catenin deletion mutant.** (A) TopFlash luciferase reporter activity induced by β-catenin* or the aromatic mutant constructs in HEK293T β-catenin KO cells. Cells were transfected with separate plasmids encoding the reporter gene and the FLAG-β-catenin mutant constructs or pcDNA3.1 as a negative control. Corresponding western blots show expression of the FLAG-β-catenin mutant constructs. Data are plotted as mean ± SD ($n = 3$). (B) Quantification of western blots from 3 independent replicate experiments for (A). (C) Micrographs of adult *Drosophila* eyes containing the indicated transgenes. (D) Quantification

of adult *Drosophila* eye area. Data are presented as mean ± SD (*n* = 8). Summary data displayed in Fig 10 can be found in S1 Data. IDR, intrinsically disordered region.

The ΔN construct is deficient in activity relative to β-catenin*. The Sept4 and SNX18 heterologous IDRs significantly rescue the degree of transcriptional activity and mutating the aromatic residues within the Sept4 and SNX18 IDRs (aroSept4 and aroSNX18) ablates activity, again in a manner reminiscent of aroN. All these constructs remove the regulatory serine and threonine residues that are phosphorylated by the destruction complex and regulate β-catenin's stability, so they are constitutively active. The expression levels of ΔN and the aromatic mutants are slightly less than β-catenin*, Sept4, and SNX18 in the experiment shown (Fig 10B). ΔN and the aromatic mutant's defect in activating TopFlash was consistently observed in experiments where they were expressed at higher levels than β-catenin* and/or their wild-type counterparts (S18 Fig). Thus, we do not believe that small differences in expression can account for the failure of ΔN and the aromatic mutants to robustly activate TopFlash.

Additionally, we made the equivalent heterologous IDR mutants in Arm and tested the activity of Sept4-Arm in the fly eye (Figs 10C and S19 for protein sequences). As seen in Fig 6A, expression of Arm* via the GMR-Gal4 driver reduces eye size as well as eye pigmentation. Removal of the N-IDR (ΔN-Arm) causes a near complete loss of these effects. Strikingly, Sept4-Arm has an intermediate effect on eye development, leading to an eye that is significantly smaller and has much less pigmentation than ΔN-Arm eyes (Fig 10D). Mutating the aromatic residues within the Sept4 IDR returns the level of Arm activity back to ΔN-Arm and wild-type levels. Our data shows that heterologous IDRs, which rescue β-catenin BMC formation in vitro, can also rescue transcriptional activity in vivo. Additionally, aromatic residues within the heterologous IDRs that are responsible for facilitating this activity, providing strong evidence for a model where the aromatic residues within β-catenin's IDRs are important for transcriptional activity through a biomolecular condensation mechanism.

## Discussion

This study provides evidence that β-catenin's ability to form BMCs is essential for its function as a transcriptional co-regulator. Mutations in the aromatic residues in the terminal IDRs of β-catenin that affect the ability to form BMCs correlate with reduced activity as a transcriptional co-regulator cultured human cells (Figs 1–5). The mutations used did not affect nuclear accumulation of β-catenin (Figs 4 and S13) or its association with TCFs and several co-activators (S10 Fig). Analogous mutations in Arm (the *Drosophila* β-catenin) also displayed signaling defects using transcriptional reporters and developmental phenotypes (Figs 6–8).

The finding that substitution of specific residues in the N-IDR (or deletion of the N-IDR) severely compromised in vitro BMC formation and in vivo signaling activity is crucial to our argument, as the N-IDR isn't traditionally thought to play a role in transcriptional activation [39,62]. Building on this result, the most compelling evidence linking BMCs to β-catenin's function involved replacing the N-IDR of β-catenin with 2 heterologous IDRs from proteins with no known role in transcription. These chimeric β-catenins rescue the deficiencies of the N-IDR deletion in BMC formation in vitro and provide significant rescue in transcriptional regulation (Figs 9 and 10). Taken together, our data provide strong support for a model where the ability of β-catenin to form BMCs is an important mechanism for its function as a transcriptional co-regulator.

The concept of BMCs playing an important role in transcriptional activation has generated a large level of support [15,16,19,63] but it is not without controversy. The Mediator subunit MED1 readily forms BMCs with Pol II subunits in vitro, and dynamic puncta containing both

complexes can be visualized on regulatory chromatin in cultured cells [64]. MED1 and co-activators such as p300 and BRD4 also form mixed BMCs with various TFs in vitro and in vivo [16,65–68]. Further studies link Pol II transcriptional bursting to a BMC model of regulatory control [19,64,69]. However, live imaging studies linking droplet formation with increased transcriptional output have produced conflicting results [70,71]. Genetic evidence linking the ability of co-activators to undergo phase separation and perform its function in transcriptional activation are limited [65,72]. There is a pressing need for further genetic studies in physiologically relevant contexts to probe the role of condensates in gene regulation.

Our work builds upon previous work [20,21] by providing an extensive functional characterization of β-catenin mutants that are deficient in BMC formation. We find that the observed deficits in β-catenin's function cannot be attributed to a defect in nuclear import (Figs 4 and S13). Considering this, the ChIP-seq data in Zamudio and colleagues [20] support a model in which β-catenin recruitment at WRE chromatin is driven by a combination of the Arm repeats (presumably due to direct binding to TCFs) and IDRs (presumably allowing β-catenin to be enriched in condensates on WRE chromatin). This model is consistent with our observations that β-catenin and LEF1 readily form heterotypic condensates in vitro (Fig 2C), while mutation of the terminal IDR aromatic residues (aroNC) reduces heterotypic condensation (Fig 2D). β-catenin lacking both IDRs, which is unable to form homotypic condensates (Fig 1) can still be recruited to LEF1 BMCs, although at a greatly reduced level compared to full-length β-catenin (Fig 2E). These results indicate that β-catenin's IDRs amplify the association of β-catenin and LEF1 driven by structured protein-protein interactions.

Our results from expressing fluorescently tagged β-catenin and LEF1 in cultured cells support our in vitro results, though there are significant differences between the 2 assays. While mutation of either the N- or C-terminal aromatic residues reduces β-catenin condensation in vitro (Figs 1B and S2A), the effect is much more dramatic in living cells (Fig 3A and 3B). Likewise, aroNC's association with LEF1 is less in cell nuclei compared to in vitro (Figs 2D and 3C). While quantitatively different, both assays are similar qualitatively, i.e., the aromatic residues of β-catenin's IDRs promote condensation with LEF1. Interestingly, expression of either LEF1 or aroLEF1 causes a dramatic nuclear accumulation of β-catenin* or aroNC (Fig 3C), which is likely due to LEF1's well-known ability to promote nuclear localization of β-catenin [73,74]. This effect may not require the formation of biomolecular condensates. Our cell culture results are consistent with recent reports demonstrating that β-catenin and TCFs co-localize in dynamic puncta [21,22]. IDRs in the central portion of TCFs, i.e., between the N-terminal β-catenin binding and HMG domains are required for condensation [22,35]. We extend those findings by demonstrating that aromatic residues within the LEF1 central IDR are required for co-localization with β-catenin (Fig 3C). It appears that aromatic residues within the IDRs of both β-catenin and LEF1 are driving co-condensation.

## Amino acid residues driving β-catenin condensate formation

One commonly proposed mechanism for BMC formation invokes pi–pi interactions between the side chain of aromatic amino acids [75]. As previously reported [20] and extended in this report, aromatic residues in the terminal IDRs of β-catenin play a key role in the ability to form condensates in vitro and in vivo. Mutation of these residues also had a context-dependent effect on β-catenin's ability to activate Wnt targets. To narrow down the number of mutations (aroN has 9, aroC has 10), we mutated subsets of aromatic residues inspired by the sticker-spacer model [40] and tested for activation of a luciferase reporter. Our data indicated that both "sticker" and "spacer" aromatic residues were crucial for β-catenin activity (S11 Fig). Comparison of pan-aromatic and conserved (between flies and humans) aromatic amino

acids in *Drosophila* developmental assays (Figs 6–8) suggest that the conserved residues (7 in the N-IDR; 5 in the C-IDR) might be the most important. In *Drosophila* transgenic assays, we compared the effect of mutating all aromatics (as with human β-catenin, 9 in the N-IDR and 10 in the C-IDR) with only the residues conserved between flies and humans (7 in the N-IDR; 5 in the C-IDR). As the pan-aromatic and conserved mutants had similar defects in signaling (Figs 6–8), it is tempting to suggest that the conserved residues contribute most to Arm activity. Further experimentation is needed to determine whether this is true for human β-catenin. The N- and C-IDRs contain a mixture of tyrosines, phenylalanines, and tryptophans (S2 and S14 Figs). BMC formation of some proteins, for example, FUS, are predominately driven by tyrosine and arginine interactions. Mutating tyrosine residues to phenylalanine strongly reduces the ability of FUS to form BMCs [76]. However, conversion of the 7 tyrosines to phenylalanine in the terminal IDRs of human β-catenin had no effect on its signaling activity (S12B Fig). This is consistent with the aromatic rings of tyrosines driving condensation rather than the hydroxyl groups.

Overall, our data are consistent with the idea that BMC formation is driven by multivalent interactions of all 3 types of aromatic residues and that many/most of the 19 residues in the IDRs contribute condensation and transcriptional activity. However, further mutagenesis is needed to test the relative contributions of each aromatic position.

Our finding that deletion of the N-IDR (ΔN) compromised β-catenin's signaling activity provided a platform on which to rescue activity with heterologous IDRs (Figs 9 and 10). But our results with ΔN are seemingly at odds with previous reports where substantial deletions in the N-IDR activate Wnt/β-catenin signaling, due to removal of phosphorylation sites necessary for ubiquitination and proteasomal degradation. For example, deletions of murine β-catenin's exon 3 (Δ5–80) activate Wnt targets in the intestine and many other tissues [77]. In *Drosophila*, the ArmS10 allele (Δ34–87) is highly active [78]. These deletions remove 6 and 5 aromatic residues, respectively, and it may be that any reduction in signaling activity is offset by stabilization. But a larger deletion (ArmΔN; Δ1–128) also has significant signaling in the fly embryonic epidermis [79]. We also see activation of Wg targets with our ΔN (Δ1–140) in some tissues, which limited our use to the TopFlash reporter assay in human cells, and the *Drosophila* eye. These results are consistent with our findings that some Wnt readouts are very sensitive to reduction in IDR aromaticity, while others had significant signaling even with aroNC, e.g., the fly epidermis and wing imaginal disc (Figs 6–8). The molecular basis of this tissue-specificity requires further experimentation.

## Is β-catenin condensation universally required for Wnt target gene activation?

Our data support an important role for β-catenin condensation in transcriptional activation, but it is unclear if this is a universal requirement for the expression of all Wnt target genes. In nearly every case we examined (the exception being aroN activating Wnt targets in HeLa cells; Fig 5A), mutation of aromatic residues resulted in a reduction of signaling activity. However, as described in the previous section, the degree of this defect depended on the assay employed. For the reporter gene assays in human cells (Figs 4 and S11) and the developing *Drosophila* eye, there was a strict requirement for aromatic residues, e.g., the 5 substitutions in aroC-con, the mutant with the fewest aromatic residue mutations, abolished Arm's signaling activity in the eye (Fig 6). For endogenous targets in HeLa cells (Fig 5) and a Wg/Wnt reporter in wing imaginal discs (Fig 7), there was an intermediate defect in signaling. In the *Drosophila* embryo, a mutant (aroNC) with 19 substitutions still had the ability to rescue a strong *arm* loss-of-function phenotype (Fig 8). It is possible that some of this rescue is the result of aroNC de-

repressing Wnt target gene expression through displacing co-repressors from TCF/Pangolin [80,81], but the modest level of expression of the aroNC transgene makes this unlikely.

Clouding the interpretation of this data is the fact that β-catenin lacking all 19 aromatic amino acids retains the ability to form BMCs in vitro at high concentrations (Fig 1). This is different from the results reported by Zamudio and colleagues, but we note that in this study, the GFP-β-catenin fusions were purified under denaturing conditions and then renatured [20]. This raises the possibility that aroNC did not properly refold. But we are also cognizant that in cultured cells, aromatic mutations do not form obvious puncta (Fig 3A and 3B). Whether this provides evidence that some Wnt readouts have a reduced requirement for condensation or that our aromatic mutants are "leaky" for BMC formation requires additional investigation. Given our results that aroNC can form BMCs in vitro (albeit only at higher concentrations), additional mutagenesis is needed to identify additional residues in the N-and C-IDRs that contribute to condensate formation. This would allow for the construction of a tighter condensate mutant to address whether BMC formation is universally required for activation of Wnt targets with increased certainty.

### Non-transcriptional BMCs containing β-catenin

In the absence of Wnt stimulation, β-catenin is targeted for proteasomal degradation by a "destruction" complex consisting of AXIN, APC, the kinases CKI and GSK3, and the E3-ubiquitin ligase β-TrCP [82]. The multivalency of the protein–protein interactions between destruction complex members led to the suggestion that it formed a BMC [82]. Indeed, the ability of AXIN to undergo phase separation has been genetically linked to efficient down-regulation of β-catenin [83] and evidence for a destruction complex BMC at endogenous levels of expression has been reported [84]. The positive Wnt effector Dvl2 has also been shown to form condensates, which has been suggested to play a role in inhibiting the destruction complex [85–87]. The aromatic β-catenin/Arm mutants described in this report also contained point mutations blocking GSK3 phosphorylation, rendering them insensitive to degradation by the destruction complex. This allowed their role in transcriptional regulation to be unambiguously assayed. It is interesting to note that when the aroNC mutation was tested with a serine at position 33 (allowing targeting for degradation by the destruction complex), we reproducibly observe higher levels of expression in wild-type β-catenin (S2B Fig). Further studies are needed to determine if this is due to decreased turnover and whether β-catenin mutants such as aroNC can be efficiently recruited to the destruction complex.

## Conclusions

Our data suggests that there is a high degree of context-dependency regarding the relationship between aromatic/condensation and activation of specific Wnt targets. Understanding the molecular basis for this specificity will require a combination of transcriptomics and an in-depth examination of WREs that have a strong requirement for aromatic residues in N-IDR and C-IDR and those that do not. Nonetheless, this report provides strong evidence that a role for β-catenin condensation needs to be considered to fully understand how the Wnt pathway activates transcription.

## Materials and methods

### Plasmids

FLAG-tagged β-catenin variants were expressed in transient transfection assays using the pCDNA3.1 vector (Thermo Fisher Scientific). A plasmid expressing human β-catenin

containing a S33Y mutation (pCDNA3-S33Y) was the starting point for further mutagenesis [88]. β-catenin mutants were created using gBLOCKS (Integrated DNA Technologies) which were subcloned into pCDNA3-S33Y that was linearized with either *BamHI* & *PmlI* (N-IDR mutants) or *BbvCI* & *XbaI* (C-IDR mutants). To express the β-catenin proteins in *E. coli*, a pET28 expression vector expressing a His-tagged GFP-β-catenin (RY8686), which was a gift from R. Young (MIT) was used [20]. Various β-catenin variants were subcloned into RY8686 using *BamHI* & *NotI*.

For the in vivo condensate assays, FLAG-eGFP-β-catenin constructs were constructed from pcDNA-FLAG-S33Y, pcDNA-FLAG-aroN, pcDNA-FLAG-aroC, and pcDNA-FLAG-aroNC constructs. These vectors were linearized with *BamHI* and the GFP insert was PCR amplified from the RY8686 construct. The pcDNA-FLAG-LEF1 vector was linearized with *EcoRI* and *NotI* and the mCherry insert was PCR amplified from the pET-His-mCherry-LEF1 vector. The linearized vectors and GFP/mCherry inserts were combined via Gibson assembly using the NEBuilder HiFi DNA Assembly Master Mix (New England Biolabs). Products were confirmed with Sanger sequencing (Genewiz, South Plainfield, New Jersey, United States of America*)*.

For the luciferase assays, the Topflash and CREAX reporter plasmids were constructed using pGL4.23 (Promega). Specific transcription factor binding sites or regulatory elements are upstream of a minimal TATA-box promoter driving the expression of the firefly luciferase gene. TopFlash has 6× TCF binding sites (plasmid was a gift from E. Fearon, University of Michigan). The CREAX luciferase reporters contain the endogenous WREs from the human *Axin2* locus [37]. For the *Defa5* reporter, the Defa5 promoter (WRE plus promixal promoter) was cloned into the promoterless pGL4.10 plasmid [41].

The pCW57.1 vector (gift from David Root, addgene plasmid #41393) was used to generate the lentiviral particles for cell transduction. In brief, the coding regions of various Flag-tagged β-catenin constructs were PCR amplified from the aforementioned pCDNA3 vectors, along with the SV40 polyadenylation site. Overlapping sequence was included to allow these amplicons to be combined with a 7.8 kB *NdeI*/*SalI* fragment of the pCW57.1 lentiviral vector via Gibson assembly using the NEBuilder HiFi DNA Assembly Master Mix (New England Biolabs). Products were confirmed with Sanger sequencing (Genewiz, South Plainfield, New Jersey, USA*)*.

Vectors for transgenic *Drosophila* strains were constructed using the PhiC31 transgenesis system [50]. An *arm* cDNA encoding 2 activating mutations (T52A and S56A) was cloned into a pUAST-FLAG-attB vector (a gift from CY Lee, University of Michigan). This construct (pUAST-Arm*-FLAG-attB) expresses the stabilized Arm protein tagged with a C-terminal FLAG epitope. Additional Arm mutants were created using gBLOCKS (Integrate DNA Technologies) cloned into pUAST-Arm*-FLAG-attB using either *MluI* & *StuI* (for N-terminal IDR mutants) or SacI & ClaI for C-terminal IDR mutants.

## Protein purification

pET28 vectors encoding the various β-catenin mutants were transformed into C41 cells and plated on LB plates containing kanamycin (50 µg/ml). Multiple fresh bacterial colonies were inoculated into 5 ml LB broth with kanamycin and grown overnight at 37˚C. The overnight cultures were diluted in 500 ml of fresh LB broth with kanamycin and grown at 37˚C until the cultures reached an $OD_{600}$ of 0.6 to 0.8. IPTG was added to a working concentration of 1 mM, and the cells were grown for 18 h at 16˚C. Cells were collected by centrifugation.

Pellets from the 500 ml cultures were resuspended in lysis buffer (50 mM Tris-HCl (pH 7.5), 300 mM NaCl, 4 mM imidazole, 0.1% Triton-X 100, 1 tablet complete protease inhibitors

(Roche, 11873580001)). Samples were then sonicated, 10 s on and 20 s off, for 3 min. Lysates were then cleared by centrifugation at 12,000×g for 12 min and added to 1 ml of pre-equilibrated TALON metal affinity resin (Takara Bio, 635502). The slurry was rotated at 4˚C for 1 h. The slurry was then centrifuged at 700 g for 5 min at 4˚C. The bead pellets were then washed twice with 10 ml of wash buffer (50 mM Tris-HCl (pH 7.5), 300 mM NaCl, 8 mM imidazole). Protein was eluted in 1.5 ml of elution buffer (50 mM Tris-HCl (pH 7.5), 300 mM NaCl, 50 mM imidazole, 10% glycerol) and the samples were rotated for 10 min at 4˚C. Protein samples were then concentrated using Amicon Ultra centrifugal filters (30 k MWCO, Millipore). The protein concentration of eluates were estimated with a Bradford assay (BioRad) and analyzed on an 8.5% acrylamide gel stained with Coomassie.

## In vitro droplet formation assay

Assays were carried out on chambered coverglass slides (Grace Bio-Labs, 112359) that were passivated with Pluronic F-127 (Sigma-Aldrich, P2243). A 5% (w/v) Pluronic F-127 solution was added to the slide's chambers and incubated for at least 1 h. After the incubation, the chambers were washed with buffer (50 mM Tris-HCl (pH 7.5), 300 mM NaCl, 10% glycerol). Varying concentrations of eGFP- and mCherry-tagged proteins were added to the chambers with droplet formation buffer (50 mM Tris-HCl (pH 7.5), 300 mM NaCl, 10% glycerol, 10% PEG-8000). The mixture was incubated for 15 min before being imaged on a Leica DMI6000B with a 63× objective and a Hamamatsu ORCA-R2 camera. Images were processed and analyzed using Leica Application Suite X (LAS X). Line profile intensities were calculated using a single line through representative condensates.

For the hexanediol sensitivity assays, 10% (w/v) 1,6-hexanediol and 2,5-hexanediol (Sigma-Aldrich, 240117 and H11904, respectively) solutions were made. These solutions were added to protein diluted in buffer (50 mM Tris-HCl (pH 7.5), 300 mM NaCl, 10% glycerol) before or after PEG-8000. All droplet assays shown were repeated at least 3 times on separate days (often with separate protein preps) and the results shown are representative.

## Cell culture, transfection, stable cell line generation, DOX treatment

HEK293T cells were obtained from the American Type Culture Collection. HeLa cells were a gift from Y. Wang (University of Michigan) and HEK293T β-catenin KO cells were a gift from K. Basler (University of Zurich) [61]. All cells were grown at 37˚C with 5% $CO_2$ and in Dulbecco's modified Eagle's medium (Gibco, 11995065) supplemented with 10% fetal bovine serum (HyClone) and penicillin-streptomycin-glutamine (Gibco, 10378016).

For transfection in HEK293T cells, 50,000 cells per well were plated in a 48-well plate and grown overnight. Cells were transfected using polyethylenimine-MAX (PEI-MAX, Poly-Sciences, 24765–1) following the manufacturer's protocol. All luciferase assays were performed at least 3 times on separate days, with similar results obtained in each experiment.

For this study, stable HeLa cell lines containing DOX-inducible β-catenin mutant (β-catenin*, aroN, aroC, and aroNC) expression cassettes were generated by lentiviral transduction. Lentiviral supernatants were made by the University of Michigan Vector core lab. To generate the mutants, HeLa cells were incubated with viral supernatants for 24 h, then the cell culture medium was replaced with fresh medium and cells were grown for an additional 24 h. Transduced cells were then selected for and maintained in cell culture medium containing 1 μg/ml puromycin. For DOX-induced expression of the mutant proteins, the individual HeLa cell lines were treated with varying doses of DOX to normalize expression across the tested mutants.

## Cell culture condensate formation assay

HEK293T β-catenin KO cells were seeded on glass coverslips in a 24-well plate and grown overnight. The following day, cells were transfected with plasmid constructs encoding GFP-β-catenin and mCherry-LEF1 using polyethylenimine-MAX (PEI-MAX, PolySciences, 24765–1), and 24 h after transfection, the cells were processed for imaging.

Coverslips in wells were washed with PBS and incubated in 4% paraformaldehyde (PFA) for 10 min. The PFA was aspirated, and cells were washed again with PBS. The cells were incubated in 0.1% PBS-T (tween-20) for 5 min and then counterstained with DAPI. Cells were washed with PBS a final time and mounted on slides with Vectashield (Vector Labs). Images were acquired with a Leica Sp8 laser confocal microscope and processed using LAS X.

## Western blotting

Cell samples were lysed and denatured in hot 1× SDS loading buffer. Protein samples were separated by SDS-PAGE and transferred to a polyvinylidene fluoride membrane (PVDF, Bio-Rad, 1620177) and blocked in 5% bovine serum albumin (BSA). Protein blots were incubated in primary antibody (diluted in 5% BSA) overnight at 4°C. After the incubation, protein blots were washed 3 times with tris-buffered saline containing 0.1% Tween-20 (TBS-T), then incubated with a secondary antibody (diluted in 5% BSA) for 1 h at room temperature. Blots were then washed 3 times with TBS-T, developed with a chemiluminescent substrate (Pierce, 34577), and imaged using a LI-COR Odyssey CLx. Images were processed in Adobe Photoshop.

Antibodies used: anti-FLAG-horseradish peroxidase (HRP, Sigma-Aldrich, A8592, 1:5,000), anti-β-tubulin (Proteintech, 66240–1, 1:20,000), anti-mouse HRP (Jackson Immu-noResearch, 115-035-003, 1:2,000).

## Immunofluorescence

HEK293T and HeLa cells were seeded on 12-mm round glass coverslips (Warner Instruments, 64–0712) in 24-well plates and grown overnight. The following day, cells were either transfected or treated with DOX to express FLAG-tagged β-catenin mutant proteins, and 24 h after transfection or DOX treatment, cells were fixed with 4% PFA (Electron Microscopy Sciences, 15710). The IF protocol was previously published (Leica, Quick guide to STED sample preparation). Briefly: following fixation, the cells were washed with PBS and permeabilized with 0.1% Triton-X 100 (MP Biomedicals, 807426). Cells were blocked with 4% BSA for 1 h, then incubated in primary antibody solution overnight at 4°C. The following morning, cells were washed and incubated in secondary antibody solution for 1 h at room temperature. Cells were then washed and counterstained with DAPI. Coverslips were mounted on slides with Vecta-shield Mounting Media (Vector Labs, H-1000). Images were acquired with a Leica Sp8 laser confocal microscope and processed using LAS X.

Antibodies used: anti-FLAG (Sigma-Aldrich, F3165, 1:1,000), anti-mouse-Alexa568 (Molecular Probes, A11031, 1:1,000), anti-Cut (Developmental Studies Hybridoma Bank, 2B10, 1:20)

## Nuclear fractionation

HEK293T and stable HeLa cell lines containing DOX-inducible β-catenin mutant (β-catenin*, aroN, aroC, and aroNC) cells were seeded on 60-mm plates and grown overnight. The following day, HEK293T cells were transiently transfected with plasmid constructs encoding FLAG-tagged β-catenin* and the aromatic mutant derivatives (aroN, aroC, and aroNC) and the stable

HeLa cell lines were treated with DOX to express the inducible proteins, and 24 h after transfection or DOX treatment, the cells were harvested for nuclear fractionation.

The REAP method was used for nuclear fractionation [89]. Briefly, cells were washed with ice-cold PBS, scraped in 1 ml of ice-cold PBS, and transferred to a pre-cooled 1.5 ml microcentrifuge tube. The cells were pelleted and resuspended in 900 μl of ice-cold 0.1% NP-40 (US Biologicals) in PBS. Cells were triturated 6 times with a P1000 micropipette, and 300 μl was transferred to a new tube as "whole cell lysate." The sample was then centrifuged and 300 μl of the supernatant was transferred to a new tube as the "cell fraction." The remaining supernatant was aspirated. The pellet was washed with 1 ml of 0.1% NP-40 in PBS and centrifuged again. The supernatant was aspirated, and the nuclear pellet was kept as the "nuclear fraction." All steps were performed in a cold room and all the equipment was pre-cooled.

## Immunoprecipitation

HEK293T β-catenin KO cells were seeded in 10-cm plates and grown overnight. The following day, cells were transiently transfected with 10 μg of plasmid constructs encoding FLAG-β-catenin*, FLAG-aroN, or FLAG-aroNC using PEI. Cells were collected 24 h after transfection for co-immunoprecipitation.

Cells were lysed in lysis buffer (50 mM Tris HCl (pH 7.4), 150 mM NaCl, 1 mM EDTA, 1% TRITON X-100, protease cocktail inhibitor (Roche)) and clarified cell lysates were analyzed by immunoprecipitation with EZview Red ANTI-FLAG M2 Affinity Gel (Millipore, F2426) beads using the manufacturer's protocol.

Antibodies used: BRG1 (ABclonal, A19556, 1:1,000), TCF1 (Cell Signaling, 2203, 1:1,000), TCF4 (Cell Signaling, 2569, 1:1,000), CBP (Cell Signaling, 7389, 1:500), p300 (Cell Signaling, 86377, 1:500), Beta-tubulin (Proteintech, 66240-I-Ig, 1:5,000), Flag (Sigma, A8592, 1:10,000).

## qRT-PCR

Total RNA was extracted using the RNeasy Plus Mini Kit (QIAGEN, 74134). cDNA synthesis was done using SuperScript III reverse transcriptase (Invitrogen, 18080–044) with oligo-dT primers. For the qRT-PCR, *Power*SYBR Green PCR Master Mix (Applied Biosystems, 4367659) was used and the reaction was carried out in a StepOnePlus Real-Time PCR System (Applied Biosystems). The β-actin and 18S genes were used as internal controls, and the relative expression of target genes was calculated using a modified Pfaffl equation which accounts for multiple reference genes [90,91]. Primer sequences listed in S1 Table. Experiments were repeated 3 times with qualitatively similar results obtained.

## Transgenic *Drosophila* strains

pUAST-Arm*-FLAG-attB and other UAS-arm derivatives were injected into M{3xP3-RFP. attP'}ZH-51C and M{3xP3-RFP.attP}ZH-86Fb embryos by BestGene (Chino Hills, California, USA) or Rainbow Transgenic Flies (Camarillo, California, USA). Transformants were identified by the presence of the mini white gene. Transgenic chromosomes were balanced over the SM5a-TM6B compound balancer, either as single inserts or in combination (51C and 86Fb). Insertions at 51C were meiotically recombined with the P[GMR-Gal4] transgene. Other Gal4 lines were obtained from the Bloomington Stock Center. Da-Gal4 and Arm-Gal4 were meiotically recombined onto a single third chromosome and balanced over TM6c. All *Drosophila* stocks were raised on yeast/glucose food and experiments were performed at 25°C, except for the Gal80$^{ts}$ experiments in Fig 7A and 7B, where cultures were raised at 18°C (to repress Gal4) and then 29°C to allow Gal4 activity for 18 h prior to fixation. In the experiments described in Fig 10C and 10D, cultures were reared at 29°C to allow maximum activity of Gal4.

## Imaging of *Drosophila* eye and wing tissues

Adult flies containing P[GMR-Gal4] and 4 copies of P[UAS-arm*] or its variants were frozen at −20˚C overnight and photographed with a Leica Stereo Dissecting Scope (Leica DMI6000B) attached to a digital camera. Eye size was quantified using ImageJ. Crosses were repeated multiple times with similar results. For Cut immunostaining, white prepupa were selected and aged 40 to 44 h at 25˚C before dissection and fixation with 4% PFA. Pupal eyes were stained with mouse anti-Cut (Developmental Studies Hybridoma Bank; 1:100). At least 10 eyes were examined for each condition, with similar results.

Wg/Wnt signaling in the wing imaginal discs was measuring using a Wnt GFP synthetic reporter previously described [55]. This reporter contains 3 copies of a grainy head binding site and 4 copies of a TCF-Helper upstream of a minimal promoter driving GFP. Larva containing this reporter and 1 copy each of P[Dpp-Gal4] and P[Tub-Gal80ts] and a single copy of P[UAS-arm*] or its variants were reared at 18˚C and then shifted to 29˚C for 18 h before selecting late third larval instar for dissection, fixation with 4% PFA and mounting. Crosses were repeated multiple times and at least 12 discs were visualized for each condition.

Quantification of fluorescent reporter activity was performed in ImageJ. A region of interest (ROI) was defined in the location of ectopic reporter activity, integrated density of the fluorescent signal was quantified, and the same ROI and calculation was used at a site of endogenous reporter activity. Data is presented as a ratio of the integrated density of the reporter at the ectopic activation site to the endogenous activation site.

To monitor expression of Arm* proteins, wing discs treated as described above and eye/ antennal discs were fixed and subjected to IF using a mouse anti-FLAG antibody (Sigma-Aldrich, F3165, 1:100) and anti-mouse-Alexa568 (Molecular Probes, A11031, 1:200) as previously described [92]. All GFP and IF images were acquired with a Leica SP8 laser confocal microscope and processed using LAS X.

## Preparation of embryonic *Drosophila* cuticles

Embryonic *Drosophila* cuticles were prepped for imaging using a previously described method [93]. Briefly, grape agar plates were added to *Drosophila* cultures for egg collection. Eggs were incubated at 25˚C and allowed to develop to the point of death, which occurs in late embryogenesis.

The embryos were dechorionated by placing them in a 50% bleach solution for 2 min, then rinsing them with distilled water, and then dried. The embryos were then devitellinized by transferring them to a 1:1 heptane to methanol solution and vigorously vortexing for 30 s. The heptane and methanol solution was decanted and the embryos were washed 3 times with methanol and a final time with a 0.1% Triton-X 100 in methanol solution. The embryos were then transferred to a glass slide, residual methanol was evaporated, and the embryos were mounted in a 1:1 solution of Hoyer's mounting medium (Hempstead Halide) and lactic acid. The embryos were imaged using a Nikon E800 upright microscope equipped with a Nikon DS-Fi3 camera and a Nikon Dark Field Condenser (Dry 0.95 to 0.80). Images were processed using NIS-Elements software.

Various P[UAS-arm*] strains were crossed with P[Da-Gal4],P[Arm-Gal4] which are both active throughout the embryonic epidermis [45,46]. Embryos contained 1 copy of each P [Gal4] and 2 copies of P[UAS-arm] transgenes for the experiments described in Fig 6. To test for rescuing activity of the different *arm* transgenes, males homozygous for P[UAS-arm] transgenes were crossed to *arm⁴*/FM7; P[Da-Gal4] females. *Arm⁴* is an amorphic allele of arm that produces a protein truncated in the sixth Arm repeat that produces no detectable protein [94]. Approximately 3/4s of the progeny displayed a consistent phenotype indistinguishable from

those of embryos with P[Da-Gal4] and the respective P[UAS-arm] construct. Approximately one quarter had a highly penetrant but distinct phenotype consistent with an $arm^4$/Y embryos with significant phenotypic rescue. All crosses were repeated multiple times; the phenotypes obtained were highly penetrant ($n > 20$ for each condition).

## Supporting information

**S1 Fig. The N- and C-terminal β-catenin IDRs are predicted to be disordered.** (**A**) IUPRED and Anchor analysis of human β-catenin. Regions with scores above 0.5 are predicted to be disordered; scores below 0.5 are predicted to be ordered. (**B**) AlphaFold prediction of β-catenin structure. The N- and C-termini lack a predicted structure, while the armadillo repeat region is α-helix rich and highly structured. (**C**) IUPRED and Anchor analysis of human LEF1. (**D**) AlphaFold prediction of LEF1 structure, indicating a mostly disordered structure. Summary data displayed in S1 Fig can be found in S1 Data.
(TIF)

**S2 Fig. Sequences of the β-catenin mutants used for in vitro droplet formation assays.** (**A**) Annotated amino acid sequences of recombinantly expressed His-eGFP-β-catenin* protein and its aromatic mutant derivatives. The specific amino acid residues that were mutated to create β-catenin*, aroN, and aroC are listed below the annotated sequence. AroNC contains all aroN and aroC mutations. (**B**) Annotated amino acid sequence of the His-eGFP-ΔNC mutant. (**C**) Annotated amino acid sequence of the His-eGFP mutant.
(TIF)

**S3 Fig. S33Y does not affect β-catenin's ability to form biomolecular condensates in vitro.** (**A**) Representative images from an in vitro droplet formation assay with the indicated mutants. Droplet assays were performed in 300 mM NaCl and 10% PEG-8000. Scale bar = 20 μm. (**B**) TopFlash luciferase reporter activity induced by S33-β-catenin (WT) or S33-aroNC (WT) constructs in HEK293T cells. Cells were transiently transfected with separate plasmids encoding the reporter genes and the β-catenin mutant constructs. Corresponding western blots show relative expression of the FLAG-β-catenin mutant constructs. Summary data displayed in S2 Fig can be found in S1 Data
(TIF)

**S4 Fig. Sequence of the LEF1 mutant used for in vitro droplet formation assays.** (**A**) Annotated amino acid sequence of recombinantly expressed His-mCherry-LEF1 protein. Aromatic residues that were changed to alanines in LEF1aro are highlighted in black. (**B**) Annotated amino acid sequence of His-mCherry.
(TIF)

**S5 Fig. Fluorescent tag controls for heterotypic in vitro droplet formation assays.** (**A**) Representative images from a heterotypic in vitro droplet formation assay with eGFP and mCherry-LEF1. The concentration of both eGFP and LEF1 protein is 8 μm. Droplet assays were performed in 300 mM NaCl and 10% PEG-8000. Scale bar = 20 μm. (**B**) Representative images from a heterotypic in vitro droplet formation assay with eGFP-β-catenin* and mCherry. The concentration of both eGFP-β-catenin* and mCherry protein is 8 μm. Droplet assays were performed in 300 mM NaCl and 10% PEG-8000. Scale bar = 20 μm. (**C**) Representative images from a heterotypic in vitro droplet formation assay with mCherry-LEF1 and the indicated GFP-β-catenin mutants. The concentration of both proteins is 8 μm. Images depict mCherry-LEF1 fluorescence only. Scale bar = 20 μm. (**D**) Violin plots depicting quantification of individual particle sizes. For β-catenin*, $n = 336$, for aroN, $n = 624$, for aroC, $n = 490$, and

for aroNC, $n$ = 900; $p$-values were calculated using one-way ANOVA followed by Dunnett's test. $^*$ = $p < 0.05$. Summary data displayed in S5 Fig can be found in S1 Data.
(TIF)

**S6 Fig. Additional line plots for β-catenin mutant and LEF1 heterotypic condensates.** (**A**) Triplicate line plots showing eGFP-β-catenin* + mCherry-LEF1. (**B**) Triplicate line plots showing eGFP-aroNC and mCherry-LEF1. (**C**) Triplicate line plots showing eGFP-ΔNC and mCherry-LEF1. White lines represent the plotted trace. Scale bar = 5 μm. Summary data displayed in S6 Fig can be found in S1 Data.
(TIF)

**S7 Fig. Diffuse eGFP-β-catenin fluorescent signal in HEK293T cells.** (**A**) Representation of diffuse eGFP-β-catenin* signal, (**B**) aroN, (**C**) aroC, and (**D**) aroNC in HEK293T cells. Scale bar = 20 μm.
(TIF)

**S8 Fig. Quantification of LEF1 and β-catenin co-localization of in HEK293T β-catenin KO cells.** (**A**) Representative single slice confocal images of HEK293T β-catenin KO cells overexpressing the indicated proteins. Scale bar = 20 μm. (**B**) Quantification of line plots showing the colocalization of eGFP-β-catenin* and mCherry-LEF1. Summary data displayed in S8 Fig can be found in S1 Data.
(TIF)

**S9 Fig. In vivo BMCs formed by FLAG-β-catenin mutants in HeLa cells.** (**A**) Representative immunofluorescence images of in vivo puncta formed by the indicated FLAG-β-catenin constructs. White arrows indicate puncta. Scale bar = 10 μm.
(TIF)

**S10 Fig. Mutating the aromatic amino acid residues in β-catenin's terminal IDRs does not affect interactions with transcription factors and transcriptional co-regulators.** (**A**) FLAG-β-catenin* and FLAG-aroN immunoprecipitates known, transcriptionally relevant binding partners. Left: total protein lysate from HEK293T β-catenin KO cells overexpressing either FLAG-β-catenin* or FLAG-aroN was used in the immunoprecipitation. Right: elution fractions were run on a gel and blotted for the indicated proteins. (**B**) Similar immunoprecipitation using total protein lysate from HEK293T β-catenin KO cells overexpressing either FLAG-β-catenin* or FLAG-aroNC.
(TIF)

**S11 Fig. A broad array of aromatics in both IDRs contribute to β-catenin activity.** (**A**) Indications of the specific amino acid residues that were mutated for each construct. (**B**) Top: Top-Flash (left) or CREAX (right) luciferase reporter activity induced by various sticker/spacer β-catenin mutants. Bottom: western blots showing the expression of each mutant construct. The protein samples used for the western blot correspond to the samples used for the luciferase assay. (**C**) Top: HD5 luciferase reporter activity induced by the sticker/spacer mutants. Bottom: corresponding western blots. Summary data displayed in S11 Fig can be found in S1 Data.
(TIF)

**S12 Fig. Mutating tyrosine residues in β-catenin's terminal IDRs does not affect protein binding or activity.** (**A**) Indications of the specific amino acid residues that were mutated for the Y-to-F construct. (**B**) Top: TopFlash luciferase reporter data induced by β-catenin* and the Y-to-F β-catenin mutant, which has all tyrosine residues in the terminal IDRs mutated to

alanines. Bottom: western blots indicating the corresponding β-catenin mutant expression. (**C**) FLAG-β-catenin* and FLAG-aroNC immunoprecipitates tyrosine phosphorylated β-catenin. Left: total protein lysate from HEK293T β-catenin KO cells overexpressing either FLAG-β-catenin* or FLAG-aroN was used in the immunoprecipitation. Right: elution fractions were run on a gel and blotted for the tyrosine phosphorylation. Summary data displayed in S12 Fig can be found in S1 Data.
(TIF)

**S13 Fig. Aromatic amino acid mutations in β-catenin's terminal IDRs does not affect nuclear localization in HeLa cells.** (**A**) Western blot showing nuclear fractionation samples from HeLa cells expressing FLAG-β-catenin* or FLAG-aroNC. (**B**) Quantification of western blots from 3 independent experiments. Data presented as the ratio of FLAG signal intensity to either β-tubulin or RNA pol II signal intensity. (**C**) Top: TopFlash luciferase reporter data induced by a dose of FLAG-β-catenin* or FLAG-aroNC expressed in HeLa cells. Bottom: western blots indicating the corresponding β-catenin mutant expression. Summary data displayed in S13 Fig can be found in S1 Data.
(TIF)

**S14 Fig. Sequences of the Arm* aromatic amino acid mutants.** (**A**) Annotated amino acid sequence of the Arm protein. All the aromatic mutants are built upon the Arm* mutation, i.e., they all have T52A and S56A.
(TIF)

**S15 Fig. Expression of Arm* and Arm mutant proteins expression in *Drosophila* embryos.** (**A**) Western blot analysis of *Drosophila* embryo lysates that were collected 4–8 h after laying. α-FLAG blot shows Arm protein expression and β-tubulin was used as a loading control. (**B**) Quantification of the ratio of the FLAG to tubulin signal intensity.
(TIF)

**S16 Fig. Effect of Arm* and aroNC expression in wild-type backgrounds.** (**A**) Representative darkfield image showing the ventral side of late embryonic *Drosophila* cuticles containing the P[Da-Gal4] and P[UAS-Arm*] transgenes. (**B**) A late embryonic *Drosophila* cuticle containing the P[Da-Gal4] and P[UAS-aroNC] transgenes.
(TIF)

**S17 Fig. Sequences of the N-terminal heterologous IDR β-catenin mutants.** (**A**) Annotated sequence of Sept4-β-catenin with indicated mutations for aroSept4-β-catenin. (**B**) Annotated sequence of Nex18-β-catenin.
(TIF)

**S18 Fig. Replicate experiments for the heterologous IDR TopFlash rescue.** (**A**) TopFlash luciferase reporter data induced by β-catenin* and the indicated heterologous IDR mutants. (**B**) Western blot indicating the expression of the various FLAG-tagged β-catenin constructs, data corresponds to panel A. Note: aroSept4 and aroSNX18 are overexpressed relative to their WT counterparts. (**C**) Additional replicate of TopFlash luciferase reporter data induced by β-catenin* and the indicated heterologous IDR mutants. (**D**) Western blot indicating the expression of the various FLAG-tagged β-catenin constructs, data corresponds to panel C. Note: ΔN is slightly overexpressed relative to β-catenin*. Summary data displayed in S18 Fig can be found in S1 Data.
(TIF)

**S19 Fig. Sequences of the N-terminal heterologous IDR Armadillo mutants.** (**A**) Annotated sequence of Sept4-Armadillo with indicated mutations for aroSept4-Armadillo. (**B**) Annotated sequence of ΔN-Armadillo.
(TIF)

**S1 Table. Sequences of primers used for qPCR.**
(XLSX)

**S1 Data. All of the data underlying the graphs shown in Figs** 2F, 2G, 2H, 3B, 4A, 4B, 4D, 4F, 5A, 5C, 6C, 7B, 10A, 10B, 10D, S1A, S2B, S5D, S6A, S6B, S6C, S8B, S11B, S11C, S12B, S13B, S13C, S18A and S18C.
(XLSX)

**S1 Raw images. All of the original blot images underlying Figs** 4A, 4E, 5B, 10A, S2B, S10A, S10B, S11B, S11C, S12B, S12C, S13A, S13C, S15A, S18B and S18D.
(PDF)

## Acknowledgments

The authors would like to thank Richard Young and his co-workers for inspiring this study, and for providing their GFP-β-catenin plasmid. Thanks to Claudio Cantù and Konrad Basler for providing the HEK293T β-catenin knockout cell line. Special thanks to Sarah Bui for assistance with lentivirus infections and line analysis of heterotypic BMCs and to Yanzhuang Wang for use of his BL2 biological safety cabinet. Thanks to Anthony Vecchiarelli for discussions on biomolecular condensates. Thanks to Hwajeong Yi, Jon Millar, Jonathan Calderon Juarez, and Carla Peralta for assistance with the construction of plasmids and to Aravind Ramakrishnan and Jon Millar for critical reading of the manuscript.

## Author Contributions

**Conceptualization:** Richard A. Stewart, Zhihao Ding, John P. Zientko, Ken M. Cadigan.

**Data curation:** Richard A. Stewart, Lauren B. Goodman, Ken M. Cadigan.

**Formal analysis:** Richard A. Stewart, Ung Seop Jeon, Lauren B. Goodman, Jeannine J. Tran, John P. Zientko, Malavika Sabu, Ken M. Cadigan.

**Funding acquisition:** Ken M. Cadigan.

**Investigation:** Richard A. Stewart, Zhihao Ding, Ung Seop Jeon, Lauren B. Goodman, Jeannine J. Tran, John P. Zientko, Malavika Sabu, Ken M. Cadigan.

**Methodology:** Richard A. Stewart, Zhihao Ding, Ung Seop Jeon, Lauren B. Goodman, Jeannine J. Tran, John P. Zientko, Malavika Sabu, Ken M. Cadigan.

**Project administration:** Ken M. Cadigan.

**Resources:** Ken M. Cadigan.

**Software:** Ken M. Cadigan.

**Supervision:** Richard A. Stewart, Ken M. Cadigan.

**Validation:** Ken M. Cadigan.

**Visualization:** Jeannine J. Tran, Ken M. Cadigan.

**Writing – original draft:** Richard A. Stewart.

**Writing – review & editing:** Richard A. Stewart, Ung Seop Jeon, Lauren B. Goodman, Jeannine J. Tran, John P. Zientko, Malavika Sabu, Ken M. Cadigan.

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
