## [Editor Report · Decision Letter 0]

9 Oct 2023

Dear Dr Cadigan, 

Thank you for submitting your manuscript entitled "beta-catenin-mediated activation of Wnt target genes utilizes a biomolecular condensate-dependent mechanism" for consideration as a Research Article by PLOS Biology.

Your manuscript has now been evaluated by the PLOS Biology editorial staff as well as by an academic editor with relevant expertise and I am writing to let you know that we would like to send your submission out for external peer review.

Once your full submission is complete, your paper will undergo a series of checks in preparation for peer review. After your manuscript has passed the checks it will be sent out for review. To provide the metadata for your submission, please Login to Editorial Manager (https://www.editorialmanager.com/pbiology) within two working days, i.e. by Oct 11 2023 11:59PM.

Kind regards,

Ines

--

Ines Alvarez-Garcia, PhD

Senior Editor

PLOS Biology

---

## [Decision Letter · Decision Letter 1]

20 Dec 2023

Dear Dr Cadigan,

Thank you for your patience while your manuscript entitled "beta-catenin-mediated activation of Wnt target genes utilizes a biomolecular condensate-dependent mechanism" was peer-reviewed at PLOS Biology. Please accept my sincere apologies for the delay in providing you with our decision. The manuscript has now been evaluated by the PLOS Biology editors, an Academic Editor with relevant expertise, and by four independent reviewers. 

As you will see, the reviewers find the conclusions novel and interesting, but they also raise several concerns that will need to be addressed to confirm the findings. Both Reviewers 1 and 2 raise several issues, including the quantification of beta-catenin levels in the mutants, which they find critical to compare them, and also quantifying the immunofluorescence images. In addition, Reviewer 1 thinks you should repeat some of the experiments in other cell lines to confirm that nuclear import of beta-catenin is not affected by aromatic mutations disrupting condensation, and Reviewer 2 would like you to show that the level of tyrosine phosphorylation of beta-catenin is not affected by the elimination of tyrosines. Reviewers 3 and 4 are more positive and mainly ask for clarifications and to show how the S33Y affects condensate formation.

In light of the reviews, we would like to invite you to revise the work to thoroughly address the reviewers' reports. Given the extent of revision needed, we cannot make a decision about publication until we have seen the revised manuscript and your response to the reviewers' comments. Your revised manuscript is likely to be sent for further evaluation by all or a subset of the reviewers.

**IMPORTANT - SUBMITTING YOUR REVISION**

3. Resubmission Checklist

a) *PLOS Data Policy*

b) *Published Peer Review*

Sincerely,

Ines

--

Ines Alvarez-Garcia, PhD

Senior Editor

PLOS Biology

Reviewers' comments

Rev. 1:

In this study, Stewart et al. shed light on the molecular requirements and functional roles of biomolecular condensate formation of the transcriptional co-factor β-catenin in the nucleus. While earlier studies pinpointed a role of the terminal intrinsically disordered regions (IDRs), how IDR their roles in mediating protein-protein interactions and/or driving condensates separately contribute to transcriptional regulation has not been addressed. The authors show that aromatic residues in both the N- and C-terminal IDR of β-catenin are important for condensate formation. As proof for the physiological relevance of biomolecular condensation of β-catenin, the authors propose that these aromatic residues are crucial for β-catenin's transcriptional role in the nucleus, while nuclear import of the protein is not affected. In addition, the authors elegantly show rescue of β-catenin condensation and transcriptional activity by replacement of its IDRs with aromatic-rich, phase separation-promoting IDRs from two unrelated proteins, strongly supporting a functional role of condensate formation. Interestingly, in vivo experiments in Drosophila indicate that the degree of dependency on β-catenin condensate formation for the activation of Wnt targets is different for different tissues, thus revealing tissue context dependency.

Overall, the manuscript is well-written and provides a clear description of the advances and limitations of the work compared to previous studies. In general, the presented work provides a very valuable contribution to the field of Wnt signaling and biomolecular condensation. Several points however will need to be addressed to validate some of the conclusions and strengthen the manuscript.

Major points:

1) The authors attempt to discriminate different aromatic β-catenin mutants with regard to their biomolecular condensation and transcriptional output. However, for many experiments quantifications of β-catenin levels are lacking. As formation of biomolecular condensates is highly dependent on protein concentration (e.g. figure 1B compare 3 vs 4 uM), it will be critical to obtain quantitative measurement of protein expression levels in the comparison of different mutants.

For instance, in figure 3A-B, the authors observe a transcriptional difference between β-catenin AroN compared to AroC and AroNC, while claiming β-catenin is expressed 'at similar levels'. Based on the shown western blot, however, aroN clearly displays higher expression levels compared to the other variants, which could explain the relatively higher transcriptional activity of this aroN mutant. Another example is shown in figure 4B, where clear differences in expression are observed upon Dox-induction, contrasting the conclusions in the text.

Similarly, most fluorescent images are not quantified. For example, in figure 4C aroN appears more highly expressed compared to other clones. Additionally, quantification of foci in this figure would be helpful, as in the current images differences in foci formation are not obvious and do not match the conclusions in the text. AroC and AroNC are claimed to not form any condensates, however small foci are visible in two out of the three images. In summary, the authors should perform quantifications for all western blots and immunofluorescent images (except for in vitro data) and weaken their claims regarding minor differences between the different β-catenin mutants, unless substantiated by equal expression levels or statistical tests.

2) To support the claim that the aromatic residues in the N-terminal IDR mainly serve to promote condensate formation required for transcriptional regulation, it would be helpful to show that the interactome of the relevant β-catenin mutants is not altered, or at least show that known interactions are not significantly altered upon mutation of the aromatic N-terminal IDR residues.

3) In Figure 3C the authors claim that nuclear import of β-catenin is not affected by aromatic mutations disrupting its biomolecular condensation. Given the novelty of this finding, the authors should substantiate this finding by repeating this experiment in other cell lines to ensure the generalizability of this observation. In addition, as stated above, quantitative measurements are required to determine the nuclear β-catenin fraction for the different mutants.

4) The authors observe clear differences between SP5 and AXIN2 transcription upon overexpression the different β-catenin mutants in Figure 4A. These results are intriguing, but puzzling. To gain better insight in the different transcriptomes of various β-catenin mutants, RNA sequencing analysis would be helpful. Alternatively, at least some of the canonical Wnt target genes, such as LGR5, CCND1 and MYC should be included in the analysis.

5) To determine the effects of the different β-catenin mutants on transcription, the authors use a doxycycline-inducible expression system. Bearing in mind the concentration-dependent formation of biomolecular condensates, it may be hypothesized that β-catenin-mediated transcription is only induced once the critical concentration for BMC formation is reached, thereby acting as an on/off switch for transcription. Can the authors use their inducible system to titrate doxycycline and determine whether indeed transcription is activated in a binary manner, or whether the response to increasing expression levels is linear?

6) Figure 4C, 3B: Both on western blot and in IF, the aroN mutant appears more stable. Potentially, the aromatic mutations in the N-terminal tail may affect β-catenin degradation, for instance by allosterically disrupting its N-terminal recognition, phosphorylation, or ubiquitination. Can the authors clarify their view on this?

7) Based on the results shown in Figure 5 and 6, where similar defects are observed for the aroN/C vs aroN/C conserved mutants, the authors conclude that the conserved aromatic residues are most important for β-catenin's function. However, in Figure S6 mutations with lower conservation (sticker and spacer mutants) also show significant signaling defects. This could suggest that mutating several aromatics could already be sufficient to compromise condensate formation. To substantiate the statement that the conserved residues are more important than the non-conserved residues, these mutants should be tested side-by-side in one of the functional assays at comparable expression levels.

Minor points:

1) To draw an easy comparison with previous work, protein concentrations should be mentioned above the microscopy images in Figure 1B, 2C-E.

2) In Figure 2H and S4C, condensates containing both LEF1 and deltaNC are shown, whereas in Figure 2E at the same concentration (middle panel) there are no condensates detected. Can the authors clarify this point?

3) Convert figure 4C and 5D to greyscale, in support of the color blind.

4) In Figure 9C and D the control and β-catenin* should be included as well.

Rev. 2:

In this manuscript, Stewart and colleagues demonstrated that aromatic amino acids of beta-catenin positioned in IDRs play a crucial role in BMC of itself using in vitro assays. Then, authors showed that these aromatic amino acids are important in forming BMC droplet with Lef-1 IDR, and this beta-catenin-Lef-1 BMC greatly accelerates Wnt signaling. Therefore, authors confirmed their findings biologically using fly models including eye development, Wnt reporter activity in wing disk, and body segmentation. The highlight of this study is that IDRs of other proteins, which are independent to Wnt signaling but are known to be BMC protein can function as well as N-IDR of beta-catenin in Wnt signal activation.

Those data authors presented here are interesting and will give us additional insights to understand how Wnt signaling is regulated by BMC. The statistical analysis, supplementary information, and data availability for this manuscript appear to be of a sufficient level. However, this manuscript in current form can be insufficient for publication in terms of experimental design, data collection, and paper writing, as described below.

Major points;

1. Overall, this manuscript requires effort and time to read. Because Figure legends contain detailed results, but the main text outlines the methodology and results. Therefore, it is necessary for understanding to go back and forth among the main text, figures, and legends frequently. Authors should explain why the experiments were selected, how the experiments were performed and what results were obtained in a way that can be understood just by reading the main text. The legend should include only explanations of the figures themselves but should not include explanations results of experiments. Please detail the results of experiments in the Results section.

2. Authors substituted all aromatic amino acids of beta-catenin IDRs into alanine to avoid BMC, resulting in Wnt signal downregulation. One of these aromatic amino acids, tyrosine is known to be not only important in BMC, but also the amino acid phosphorylated. Therefore, authors should show that the level of tyrosine phosphorylation of beta-catenin is not affected by the elimination of tyrosines. Authors can show this point by beta-catenin IP-Phospho tyrosine IB experiment. In addition, substitution of all those IDR tyrosines into other aromatic amino acids like F or W for TopFlash reporter assay may support the present data.

3. As described in the Discussion section, authors introduced S33Y mutation to stabilize beta-catenin for focusing the transcriptional activity by beta-catenin-Lef-1 BMC. Of course, we know this substitution is sometimes used for beta-catenin stabilization. However, it seems strange to employ the S-Y substitution in this study. Because authors eliminated aromatic amino acids including tyrosine for BMC but introduced another "aromatic Y" in same IDR for protein stabilization. I wondered why authors did not introduce S33A mutation as in many other studies and in this study (Y-A, F-A, W-A). Authors should show 'no difference' in droplet formation and TopFlash assays between S33Y- and S33A-based beta-catenin mutants.

4. In Fig. 2, authors introduced substitutions of aromatic amino acids only in beta-catenin IDRs to confirming BMC mimicking interaction between beta-catenin and Lef-1. Author should additionally show the results when aromatic substitutions are introduced in Lef-1 IDR.

5. Authors showed the co-localization/interaction of beta-catenin and Lef-1 in BMC mimicking droplet in vitro in Fig. 2C and localization of beta-catenin in nucleus in Fig. 3C. Authors should show the colocalization of beta-catenin and Lef-1 in puncta observed in nucleus.

6. In Fig. 3C and D, authors compared the level of nuclear beta-catenin by immunofluorescent experiments. Authors should show more quantitative data. The level of nuclear beta-catenin can be evaluated by WB experiment with anti-beta-catenin after the separation of nuclear fraction.

7. Authors showed in Fig. 9A that IDRs of Sept4 and SNX18 can be functionally replaced with N-IDR of beta-catenin and loss of aromatic amino acids of Sept4 impaired beta-catenin function in Wnt signaling. Please include the results also with the loss of aromatic amino acids of SNX18 IDR.

8. Authors can combine Fig. 6 with 7.

Minor points;

1. In the legend of Fig. 1. We could not see the indication of (B).

2. In Fig. 7E, Da::Arm* should be Da::arm*.

3. Authors did not describe anything for Fig. 8C in the main text.

I fell that all of the above is required for the publication in PLoS Biology except Major point #8.

Rev. 3: Mark Peifer – note that this reviewer has signed his review.

Transcriptional regulation by the Wnt/beta-catenin pathway is key to development and tissue homeostasis and is disrupted in disease. Many transcription factors contain intrinsically disordered regions that can mediate phase separation in vitro, but the roles of these in vivo remain less clear, as few in vivo studies have taken apart the IDRs and assessed function. Here the Cadigan lab builds on an earlier observation that beta-catenin can phase separate in vitro and in vivo and that began assessment of the roles of aromatic residues. They substantially extend these earlier observations—characterizing in detail the roles of both the N-terminal and C-terminal IDRs and of the conserved and less conserved aromatic residues in each, using multiple assays. In an impressive series of experiments, they assess phase separation in vitro, transcriptional activity on reporter genes and on endogenous targets in mammalian cells, and, most impressive, signaling activity in vivo in several contexts in Drosophila. The data are carefully quantified and support their conclusions. These data lead to a much more nuanced picture and provide one of the most complete analyses of the IDRs of a transcription factor in vivo. This work will be of broad interests to cell and developmental biologists. I had some suggestions about clarity and a few questions about the data presented that should be straightforward to address.

1. There was one important deficit in the discussion of prior work. Many of the aromatic residues in the N-terminal IDR (and 6 of 8 conserved aromatic residues) are deleted in the ArmS10 mutant, which retains strong signaling activity-- many of the authors' effects on genuine in vivo targets support the idea that such a mutant should retain significant activity. It's also worth noting that the same paper suggests Arm∆N, deleting all N-terminal amino acids, is less active than ArmS10. In earlier work from the same lab, the authors found that ArmS14, which deletes the last two conserved aromatic acids in the N-terminal domain (those not deleted by ArmS10) is fully active in signaling in vivo.

2. Fig. 2B vc C. Did beta-catenin enhance droplet formation by Lef1. It appears so, and if so this is worth noting.

3. The authors should tell us in the main text the nature of the amino acid change(s) in Arm* and define in the text how aroN and aroC differ from aroNcon and aroC-cons

4. The authors say: "All the aromatic Arm mutants (which contain the stabilizing mutants of Arm*) exhibit minimal signaling activity in this assay". This needs to be more nuanced since they actually show they have low but statistically significant signaling activity.

5. The authors should move Fig S8A into the main figure as it provides important information.

6. The assay is Figure 7 is the most interesting of all of their assays, as it integrates effects on all target genes in vivo. It provides important information about ArmNC function. Did they test any of their other mutants in the Fig. 7 assay?

7. Fig 8. It is worth noting that that aroSept4 did rescue spherical condensates, though not at the same concentration

8. The chimeras remove the residues that are involved in beta-catenin phosphorylation and targeting for destruction—this needs to be noted in these experiments.

9. The authors stated "The finding that substitution of specific residues in the N-IDR (or deletion of the N-IDR) severely compromised BMC formation and signaling activity"—this needs to be nuanced to make it clear this was only true IN VITRO.

Minor issues

Fig S1 It is worth noting that AlphaFold predicts alpha-helical regions in the N-terminal IDR (one of which is the alpha-catenin binding site )

Rev. 4:

In the manuscript titeled "Beta-catenin-mediated activation of Wnt target genes utilizs a biomolecular condensate-dependent mechanism" the authors explore the contribution of beta-catenin phase-separation to it's capacity to drive transcription. The authors perform elegant and detailed mutational analysis and use in vitro, cell culture and fly in vivo models to evaluate the contribution of aromatic amino acids to beta-catenin condensate formation and transactivation. These findings are timely and build upon earlier work by Zamudio et al., as noted by the authors. This work supports a link between condensate formation and active transcription, but also inform that there is condensate-independent transactivation by beta-catenin. The work is generally of high quality and provides novel insights in condensate formation and transcriptional activation. I only have one minor comment.

Fig 1. How does the S33Y affect condensate formation? Could it be the explanation of the difference between Zamudio et al and this report? It adds one aromatic Y to the IDR, therefore the aroN is not fully mutated. In light of the aromatic IDR substitution data, and the reduced activity of Arm with as little as 5 aromatic substitutions, this could be plausible explanation. Can the authors test a S33C or S33P oncogenic mutant and see if the residual in-vitro droplet formation is maintained?

---

## [Decision Letter · Decision Letter 2]

31 Jul 2024

Dear Dr Cadigan,

Thank you for your patience while we considered your revised manuscript entitled "beta-catenin-mediated activation of Wnt target genes utilizes a biomolecular condensate-dependent mechanism" for publication as a Research Article at PLOS Biology. This revised version of your manuscript has been evaluated by the PLOS Biology editors, the Academic Editor and three of the original reviewers.

Based on the reviews, we are likely to accept this manuscript for publication, provided you satisfactorily address the remaining points raised by Reviewer 2. Please also make sure to address the data and other policy-related requests stated below.

In addition, we would like you to consider a suggestion to improve the title:

"Wnt target gene activation requires the phase separation of beta-catenin into biomolecular condensates”

We expect to receive your revised manuscript within two weeks. 

*Published Peer Review History*

*Press*

Sincerely,

Ines

--

Ines Alvarez-Garcia, PhD

Senior Editor

PLOS Biology

DATA POLICY: PLEASE READ

Fig. 2F-H; Fig. 3B; Fig. 4A, B, D, F; Fig. 5A, C; Fig. 6C; Fig. 7B; Fig. 10A, B, D; Fig. S1A, C; Fig. S3B; Fig. S5D; Fig. S6A-C; Fig. S7B; Fig. S9B, C; Fig. S10B; Fig. S11B, C and Fig. S16A, C

CODE POLICY

We require the original, uncropped and minimally adjusted images supporting all blot and gel results reported in an article's figures or Supporting Information files. We will require these files before a manuscript can be accepted so please prepare and upload them now. Please carefully read our guidelines for how to prepare and upload this data: https://journals.plos.org/plosbiology/s/figures#loc-blot-and-gel-reporting-requirements

Reviewers' comments

Rev. 1:

The authors have addressed all my concerns. The manuscript has greatly improved, compliments to the authors for their achievements. The work strengthens the connection of Wnt signaling and biomolecular condensates and will be of broad interest to researchers working in areas of cell and developmental biology. I will be more than happy to support publication of this work.

Rev. 2:

The authors responded sincerely to many of the issues raised and provided numerous pieces of evidence to support the story. In this revised format, almost all the data except one was of a high enough standard to be published. Now the paper excellently presented important new findings.

However, it seems that only the data on the localization of nuclear β-catenin may still be confusing to readers. In this manuscript, Fig. 3A and B show that nuclear BMC puncta are observed only when IDRs are intact, but when IDRs are lost, BMC puncta are not formed. In these panels, almost no signal of β-catenin itself that has lost IDRs is observed in the nucleus surrounded by white dashes despite the presence of nuclear β-catenin(aroN/C) protein showing in other figures. Why?

On the other hand, Fig. 3C and Fig. 4E clearly show by imaging and biochemistry that the presence or absence of IDRs does not affect the nuclear localization of β-catenin, strongly supporting the authors' claims. In addition, Fig. 4D also shows that the nuclear localization of β-catenin is not affected by IDRs. Fig. 4D also quantitatively shows that the nuclear localization of β-catenin is not affected by the presence or absence of IDRs, and therefore supports 3C and 4E, since the same amount of β-catenin is present in the nucleus regardless of the presence or absence of IDRs.

However, in this figure (4C), which is the basis for Fig. 4D, the BMC puncta formation in β-catenin* is unclear, making it difficult to make a qualitative assessment. Is it possible to adjust the contrast of the image a little more so that only β-catenin* clearly forms BMC puncta with same amount of nuclear β-catenin proteins?

All except for the problem above are consistent, and I believe this is an excellent paper worthy of acceptance. Congratulations.

Rev. 3: Mark Peifer

Transcriptional regulation by the Wnt/beta-catenin pathway is key to development and tissue homeostasis and is disrupted in disease. Many transcription factors contain intrinsically disordered regions that can mediate phase separation in vitro, but the roles of these in vivo remain less clear, as few in vivo studies have taken apart the IDRs and assessed function. Here the Cadigan lab builds on an earlier observation that beta-catenin can phase separate in vitro and in vivo and that began assessment of the roles of aroma1c residues. They substantially extend these earlier observations—characterizing in detail the roles of both the N-terminal and C-terminal IDRs and of the conserved and less conserved aroma1c residues in each, using multiple assays. In an impressive series of experiments, they assess phase separa1on in vitro, transcriptional activity on reporter genes and on endogenous targets in mammalian cells, and, most impressive, signaling ac1vity in vivo in several contexts in Drosophila. The data are carefully quantified and support their conclusions. These data lead to a much more nuanced picture and provide one of the most complete analyses of the IDRs of a transcription factor in vivo. I had some concerns about the original manuscript and they have fully addressed them. In my opinion, the new data they added also addresses the concerns of the other reviewers. This work significantly advances our field, and will be of broad interests to cell and developmental biologists.

---

## [Editor Report · Decision Letter 3]

30 Aug 2024

Dear Dr Cadigan,

Thank you for the submission of your revised Research Article entitled "Wnt target gene activation requires beta-catenin separation into biomolecular condensates" for publication in PLOS Biology. On behalf of my colleagues and the Academic Editor, Bon-Kyoung Koo, I am delighted to let you know that we can in principle accept your manuscript for publication, provided you address any remaining formatting and reporting issues. These will be detailed in an email you should receive within 2-3 business days from our colleagues in the journal operations team; no action is required from you until then. Please note that we will not be able to formally accept your manuscript and schedule it for publication until you have completed any requested changes.

PRESS

Sincerely, 

Ines

--

Ines Alvarez-Garcia, PhD

Senior Editor

PLOS Biology
